# Engineered *Mycobacterium tuberculosis* triple-kill-switch strain provides controlled tuberculosis infection in animal models

Xin Wang [1,10], Hongwei Su[2,9,10], Joshua B. Wallach[2,10], Jeffrey C. Wagner[1,10], Benjamin J. Braunecker[1], Michelle Gardner[1], Kristine M. Guinn[1], Nicole C. Howard[1], Thais Klevorn [2], Kan Lin [2], Yue J. Liu[1], Yao Liu [2], Douaa Mugahid[1], Mark Rodgers [3,4], Jaimie Sixsmith[1], Shoko Wakabayashi[1], Junhao Zhu[1], Matthew Zimmerman[5], Véronique Dartois [5,6], JoAnne L. Flynn [3,4], Philana Ling Lin[4,7], Sabine Ehrt [2]✉, Sarah M. Fortune [1,8]✉, Eric J. Rubin [1]✉ & Dirk Schnappinger [2]✉

Human challenge experiments could accelerate tuberculosis vaccine development. This requires a safe *Mycobacterium tuberculosis* (Mtb) strain that can both replicate in the host and be reliably cleared. Here we genetically engineered Mtb strains encoding up to three kill switches: two mycobacteriophage lysin operons negatively regulated by tetracycline and a degron domain–NadE fusion, which induces ClpC1-dependent degradation of the essential enzyme NadE, negatively regulated by trimethoprim. The triple-kill-switch (TKS) strain showed similar growth kinetics and antibiotic susceptibilities to wild-type Mtb under permissive conditions but was rapidly killed in vitro without trimethoprim and doxycycline. It established infection in mice receiving antibiotics but was rapidly cleared upon cessation of treatment, and no relapse was observed in infected severe combined immunodeficiency mice or Rag$^{-/-}$ mice. The TKS strain had an escape mutation rate of less than $10^{-10}$ per genome per generation. These findings suggest that the TKS strain could be a safe, effective candidate for a human challenge model.

Tuberculosis (TB) stands as a leading global infectious threat, with its toll on lives and health exacerbated during the coronavirus disease 2019 pandemic. Mathematical modelling underscores the urgency of developing novel TB vaccines to achieve the ambitious goals set by the 'Global Plan to End TB' initiative by Stop TB Partnership[1]. The only licensed TB vaccine remains the attenuated *Mycobacterium bovis* strain known as Bacille Calmette–Guerin (BCG). Despite its efficacy against TB meningitis and miliary TB in infants[2], BCG's effectiveness against TB in adolescents and adults varies geographically[3]. An alternative vaccine candidate, M72/AS01E (ref. 4), exhibited protection rates against active TB no higher than 50% in adults with latent *Mycobacterium tuberculosis* (Mtb) infection, considerably lower than the 90% efficacy seen with polio and measles vaccines[5,6]. The challenges hindering TB vaccine development are twofold: (1) the absence of inexpensive and predictive preclinical animal models and (2) the lack of validated immunological markers for protection. While progress has been made in understanding immune protection indicators[7–10], leveraging non-human primates (NHPs) as models, a controlled human infection model (CHIM), if it can be developed, would offer a valuable tool to assess vaccine efficacy and complement animal studies.

A full list of affiliations appears at the end of the paper. ✉e-mail: sae2004@med.cornell.edu; sfortune@hsph.harvard.edu; erubin@hsph.harvard.edu; dis2003@med.cornell.edu

The CHIM, also referred to as the human challenge model, is a pivotal instrument in advancing vaccine development and gauging treatment efficacy. In this strategy, volunteers are deliberately infected with an infectious agent to evaluate the efficacy of vaccines or therapeutic candidates. Human challenge studies possess advantages not reproducible in natural infection investigations. By eliminating variables such as diverse genetic backgrounds of infectious agents, variable infectious doses, uncertain exposure periods and patient comorbidities, researchers can pinpoint protective host factors and immediate responses during infection. CHIM facilitates meticulous control over infection rates and timing, enabling comparisons among new vaccine candidates, modified regimens and preventive, preemptive or postsymptomatic treatments. Its success includes the development of various Food and Drug Administration-approved vaccines and therapies, such as the cholera vaccine Vaxchora[11], the influenza therapeutic Oseltamivir[12], the Vi-tetanus toxoid conjugate vaccine for *Salmonella typhi*[13] and dosing schedules and adjuvant selections for malaria vaccines RTS,S and S/AS01 (refs. 14,15). A recent BCG human challenge model conducted in South Africa introduced live BCG or purified protein derivative (PPD) into lung segments of volunteers through bronchoscopy, followed by immune profiling studies[16]. The observed adverse effects were mild, marking an important step towards establishing a CHIM in the realm of TB research.

A successful CHIM study must align with the ethical principle of *primum non nocere*[17,18]. For TB CHIM, this requires engineering of an attenuated Mtb strain that adheres to preclinical safety standards. This challenge strain needs to be capable of replication in the naive host to ensure that its eradication requires vaccine-boosted immune responses. Moreover, the challenge strain should permit only a limited window for replication to adhere to safety regulations, with the genetic circuit governing strain mortality demonstrating a low escape mutation rate (we set a stringent goal of $<10^{-12}$ per genome per generation). To attain this objective, we designed and integrated three distinct kill switches employing orthogonal mechanisms into the Mtb strain H37Rv. The resultant triple-kill-switch (TKS) strain exhibited a growth rate and antibiotic susceptibility similar to the wild-type H37Rv under permissive conditions and was rapidly killed both in vitro and within mouse models under restrictive conditions. We determined that the escape rate for all three kill switches remained below our current limit of detection ($<10^{-10}$ per genome per generation). Importantly, the TKS strain did not induce relapse in mice with severe combined immunodeficiency (SCID) and Rag$^{-/-}$ mice. Based on our safety and efficacy assessments, we proposed the TKS strain as a promising candidate for a human challenge strain.

## Results

### Construction of tetracycline addicted dual-lysin Mtb
We constructed two genetic switches that depend on a tetracycline, such as anhydrotetracycline (aTc) or doxycycline (hereafter referred to as doxy), to repress the lysin operons, L5L and D29L, of two mycobacteriophages, L5 and D29. L5L was repressed by a single-chain reverse TetR (revTetR)[19] (Fig. 1a). D29L was repressed by a modified TetPipOFF system[20] in which a single-chain wild-type TetR represses PipR, which in turn controls activity of the D29L operon. To ensure independent regulation of the lysin operons, the two TetRs chosen were distinct in their operator specificity (revTetR binds to the operator mutant $tetO_{4C5G}$ while TetR binds to wild-type $tetO$) and in their allosteric response to aTc/doxy (revTetR requires aTc/doxy to be active, whereas TetR is inactivated by aTc/doxy). Single-chain repressors were used to prevent heterodimerization of the two TetRs[21]. The resulting 'dual-lysin strain' is a derivative of Mtb H37Rv that carries both switches integrated in its genome.

In an axenic culture, the dual-lysin strain exhibited growth in the presence of aTc, while in its absence, it displayed potent bactericidal effects, with a kinetic rate of a ~1 log reduction in colony-forming unit

(CFU) every 3 days (Fig. 1b). Through fluctuation assays, we determined the in vitro escape rate as $8.8 × 10^{-9}$ per genome per generation (Fig. 1c). We mapped escape mutations to both lysin kill-switch elements via whole-genome sequencing. In the L5L kill switch, most mutations were mapped to the *tetO* promoter region and lysin A. This indicates the L5L escape arose from either disruption of the revTetR regulation or compromised lysin A function. In the D29L kill switch, frameshift and deletion mutations were found in *tetR-tetO* elements, indicating that escape of D29L kill switch is probably due to alteration in TetR function (Fig. 1d and Supplementary Table 1). In a C57BL/6 murine model, the dual-lysin strain effectively established an infection and persisted as long as doxy was administered. Following doxy withdrawal, the strain was cleared from both lungs and spleen with killing kinetics of approximately 0.50 log per week and 0.35 log per week, respectively (Fig. 1e,f). In NHPs (here Mauritian cynomolgus macaques) using the Mtb Erdman strain with the dual-lysin constructs, the NHPs were kept on either a daily doxy regimen for 6 weeks post Mtb challenge or a regimen of 2 weeks of daily doxy, followed by 4 weeks of a regular diet post Mtb challenge. All six NHPs were subjected to necropsies at week 6 post challenge to evaluate bacterial burden (Fig. 1g). A 6-week doxy regimen resulted in infection reaching up to $10^6$ CFU in the thorax, lungs and spleen. Shortening the doxy supplementation to 2 weeks led to sterilization of all assessed organs, except for a solitary instance of $10^3$ CFU in the thorax and lungs of one animal (Fig. 1h,i). Notably, the presence of viable colonies in the one NHP under the 2-week doxy treatment was not attributed to escape mutations, as we verified that bacterial growth was still regulated by aTc in axenic culture. Lung granulomas were only found in one of the four macaques on the 2-week doxy regimen, while both macaques on doxy for the duration of the study had visible lung granulomas (Fig. 1l). Similarly, none of the 2-week doxy regimen macaques had thoracic lymph nodes with signs of disease, while both 6-week doxy regimen macaques had involved lymph nodes (Fig. 1m).

Collectively, these findings underscore the stringent regulation of the dual-lysin strain's viability by aTc, both in vitro and in vivo. However, the kinetics of bacterial elimination were not as swift as desired, and the escape rate surpassed our benchmark, thereby raising safety concerns regarding the strain's suitability for human challenge studies. To achieve the target safety standards, our focus shifted towards developing a third kill switch.

### Engineering TMP-regulated Mtb proteolysis
To design a killing mechanism orthogonal to lysin expression, we sought to leverage degradation of essential and vulnerable Mtb proteins. Prior research has described a degron engineered to mediate protein degradation in mammalian central nervous system[22]. This degron was characterized by a mutated *Escherichia coli* dihydrofolate reductase that forms an unstable protein conformation, consequently driving protein degradation. In the context of eukaryotic system, proteolysis was hindered by trimethoprim (TMP), an ineffective antibiotic against Mtb[23,24], which binds to the degron (ddTMP), to stabilize its unfolded conformation. To investigate the applicability of the ddTMP degron in mycobacteria, we fused a red fluorescent protein (RFP; mCherry in *Mycobacterium smegmatis* (Msm) or mScarlet in Mtb) with ddTMP at both the C- and N-termini and evaluated the TMP-dependent stability of the resultant RFP construct in Msm. In the absence of TMP, RFP was susceptible to proteolysis when ddTMP was fused at either the N- or C-terminus, but the presence of TMP prevented degradation of RFP (Fig. 2a–c). In Mtb, the ddTMP-RFP was similarly stabilized in a TMP-dependent manner (Fig. 2d).

To decipher whether the ddTMP-mediated proteolysis was dependent on ClpC1 or ClpX acting as the ATPase with the ClpP protease, we analysed the degradation kinetics of ddTMP-RFP in aTc-induced *clpC1*- or *clpX*-knockdown Msm strains. Our data indicate that depleting ClpC1 led to the accumulation of ddTMP-RFP. Conversely, the abundance of ddTMP-RFP was unaffected by the depletion

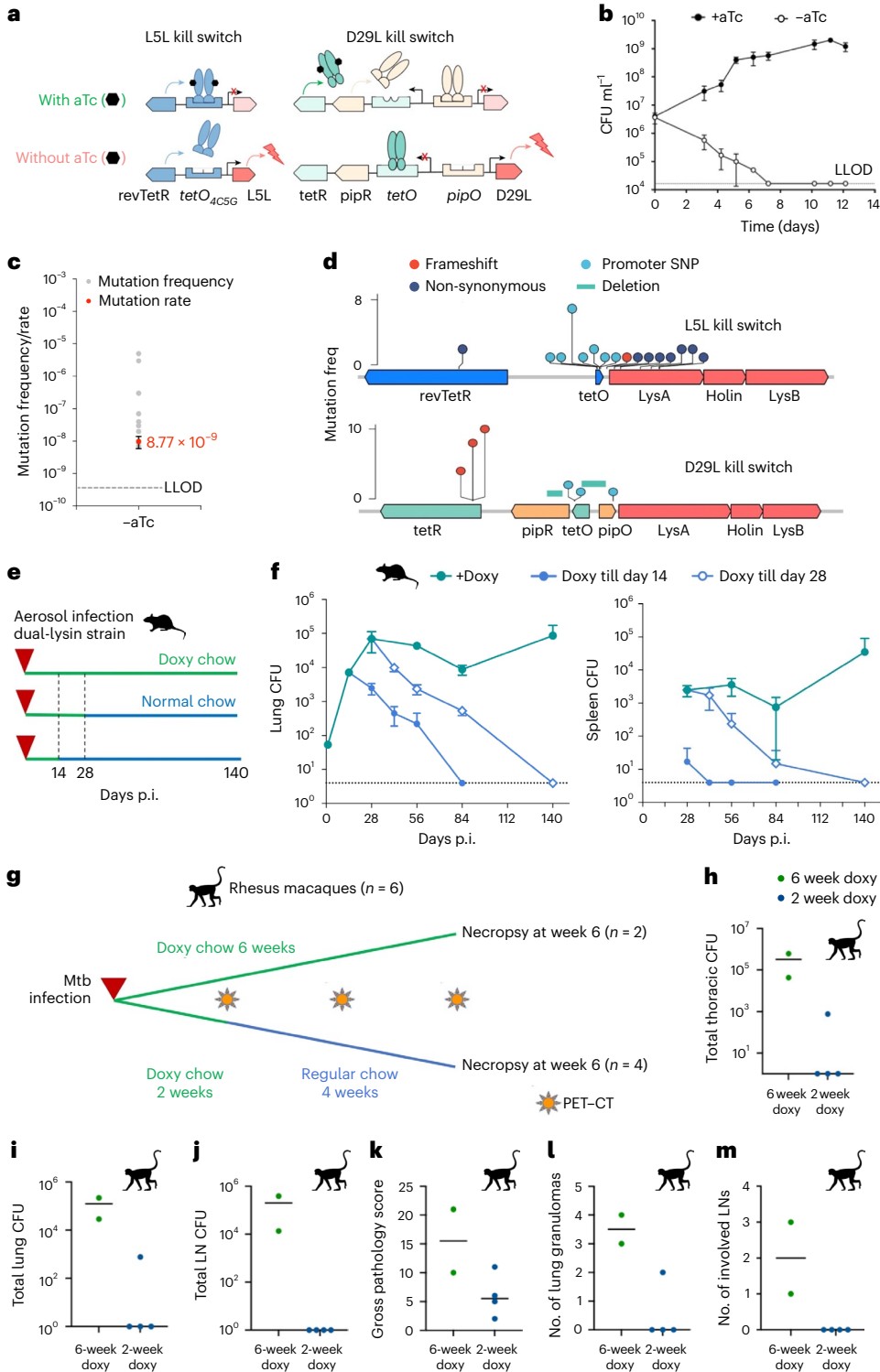

**Fig. 1 | The dual-lysin kill switch is effectively bactericidal in Mtb both in vitro and in vivo. a**, Regulatory scheme of aTc-inducible lysin expression. **b**, The kill curve of dual-lysin Mtb strain in vitro. Each dot shows the mean bacterial burden from biological triplicates. The error bars represent the standard deviation. The dashed line shows the CFU lower limit of detection (LLOD). **c**, The escape mutation rate of dual-lysin Mtb strain in the absence of aTc. The mutation rate was calculated from escape frequency of 20 independent cultures. The escape frequency is presented as the mean, with the error bar showing the standard deviation. **d**, The escape density plot shows mutations in L5-lysin kill switch and D29-lysin kill switch from 20 independent escape mutants. The mutation type and frequency in each individual kill switch are noted. **e**, The scheme of C57BL/6 mouse infection with Mtb dual-lysin strain. p.i., post infection.

**f**, The clearance kinetics of Mtb dual-lysin strain in C57BL/6 mice with doxy depleted at day 14 or day 28. Each symbol shows the mean bacterial burden from five mice. The error bars represent the standard deviation. The dashed line shows the CFU LLOD. **g**, The scheme of doxy-dependent NHP infection with the dual-lysin kill-switch strain. **h**–**m**, The NHP infection outcome of the dual-lysin Mtb strain supplemented with doxy for 2 or 6 weeks; the outcome is judged by total thoracic CFU (**h**), lung CFU (**i**), lymph node (LN) CFU (**j**), gross pathology score (**k**), number of lung granulomas at necropsy (**l**) and number of involved LNs (**m**). Each symbol represents a macaque, and the line represents the median. Panels **e**–**m** created with BioRender.com, released under a Creative Commons Attribution-NonCommercial-NoDerivs 4.0 International license.

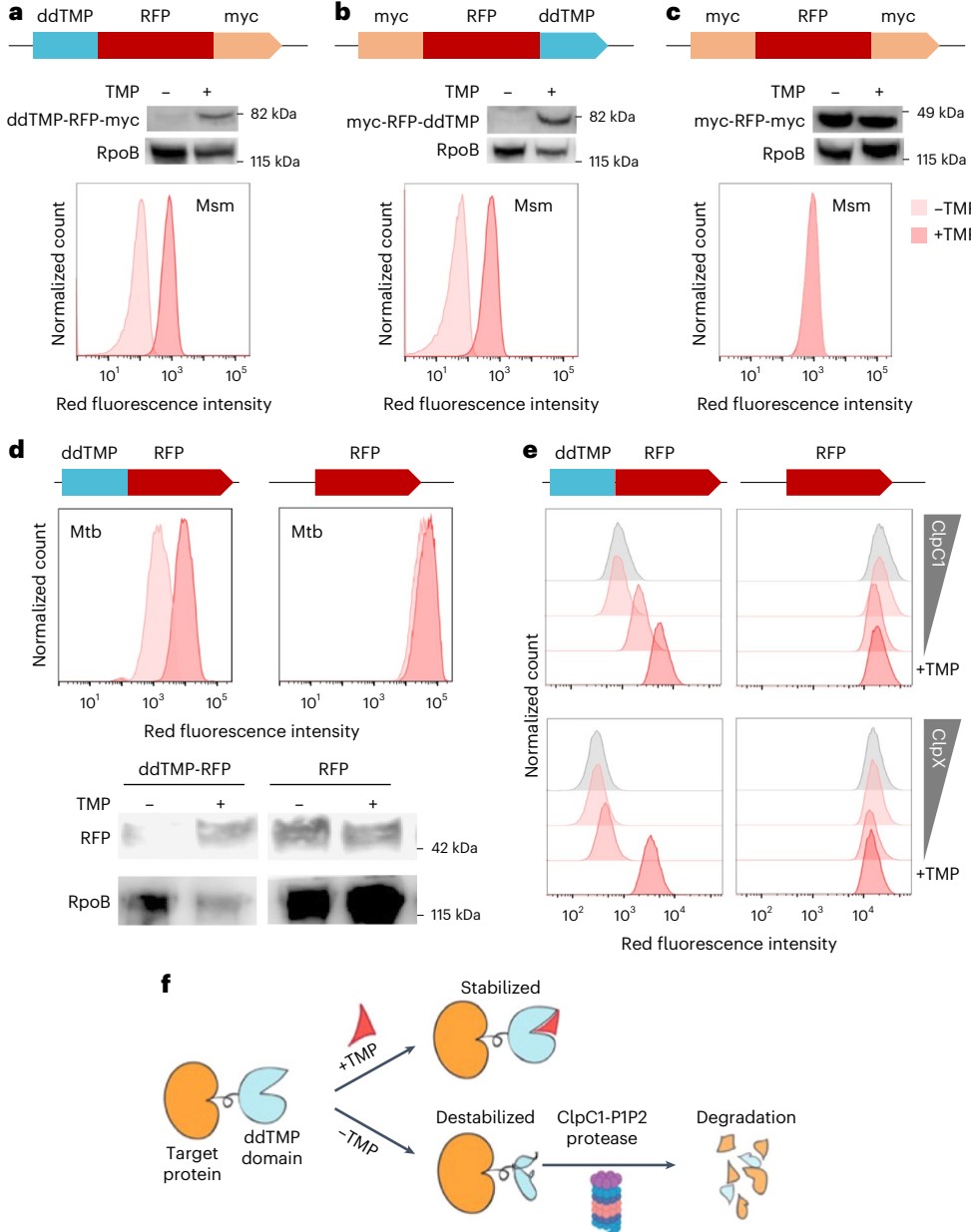

**Fig. 2 | Mycobacteria proteins fused with a ddTMP tag are subjected to TMP-dependent degradation via ClpC1-ClpP1P2 protease. a–c**, mCherry fused with ddTMP at the N-terminus (**a**), the C-terminus (**b**) or null (**c**). The ddTMP at either the N- or C-terminus manifests TMP-dependent degradation in Msm. **d**, ddTMP–mScarlet shows a TMP-dependent degradation in Mtb. **e**, Flow cytometry with *clpC1*- or *clpX*-knockout Msm indicates that the TMP-dependent degradation is mediated via protease ClpC1-ClpP1P2. The bottom row shows the RFP steady state measured in mid-log culture supplied with TMP. *ClpX* or *clpC1* were knocked down via a gradient of aTc in the absence of TMP. **f**, The ClpC1-dependent degradation mechanism of the ddTMP tag in mycobacteria. All experiments have been repeated independently three times and show the representative results. Panel **f** created with BioRender.com, released under a Creative Commons Attribution-NonCommercial-NoDerivs 4.0 International license.

of ClpX (Fig. 2e). These data indicate that ddTMP-tagged proteins undergo degradation primarily through ClpC1 as the cognate ATPase with the Clp protease system in Msm (Fig. 2f).

### TMP-dependent Mtb NadE degradation is bactericidal

We sought to integrate the ddTMP degron into Mtb to engineer a kill-switch strain. We fused the ddTMP degron to the C-terminus of the essential Mtb protein NadE at its native chromosomal locus (Fig. 3a). NadE is essential for catalysing the final step in NAD$^+$ de novo biosynthesis. NadE depletion has been shown to induce rapid killing during both replication and persistence[19] as a result of the reduction of both NADPH and NADH pools[25]. We assessed the kinetics of NadE degradation using a reporter construct where NadE was fused with mScarlet-ddTMP at the C-terminus. Absent external TMP, we observed exponential degradation of NadE-mScarlet-ddTMP, with a half-life of ~1 h (Fig. 3b). The minimum concentration of TMP required to support the growth of the NadE-ddTMP strain was 3 ng ml$^{-1}$ in axenic culture. At TMP concentrations higher than 30 ng ml$^{-1}$, the NadE-ddTMP strain displayed no apparent growth defect compared with the wild-type H37Rv strain (Fig. 3c). In the absence of TMP supplementation, the degradation of NadE had a bactericidal effect, evident in a 2 log reduction in CFU within a 12-day period in vitro (Fig. 3d).

Since we observed TMP-regulated growth and killing, we assessed the escape mutation rate and characterized the mutations leading

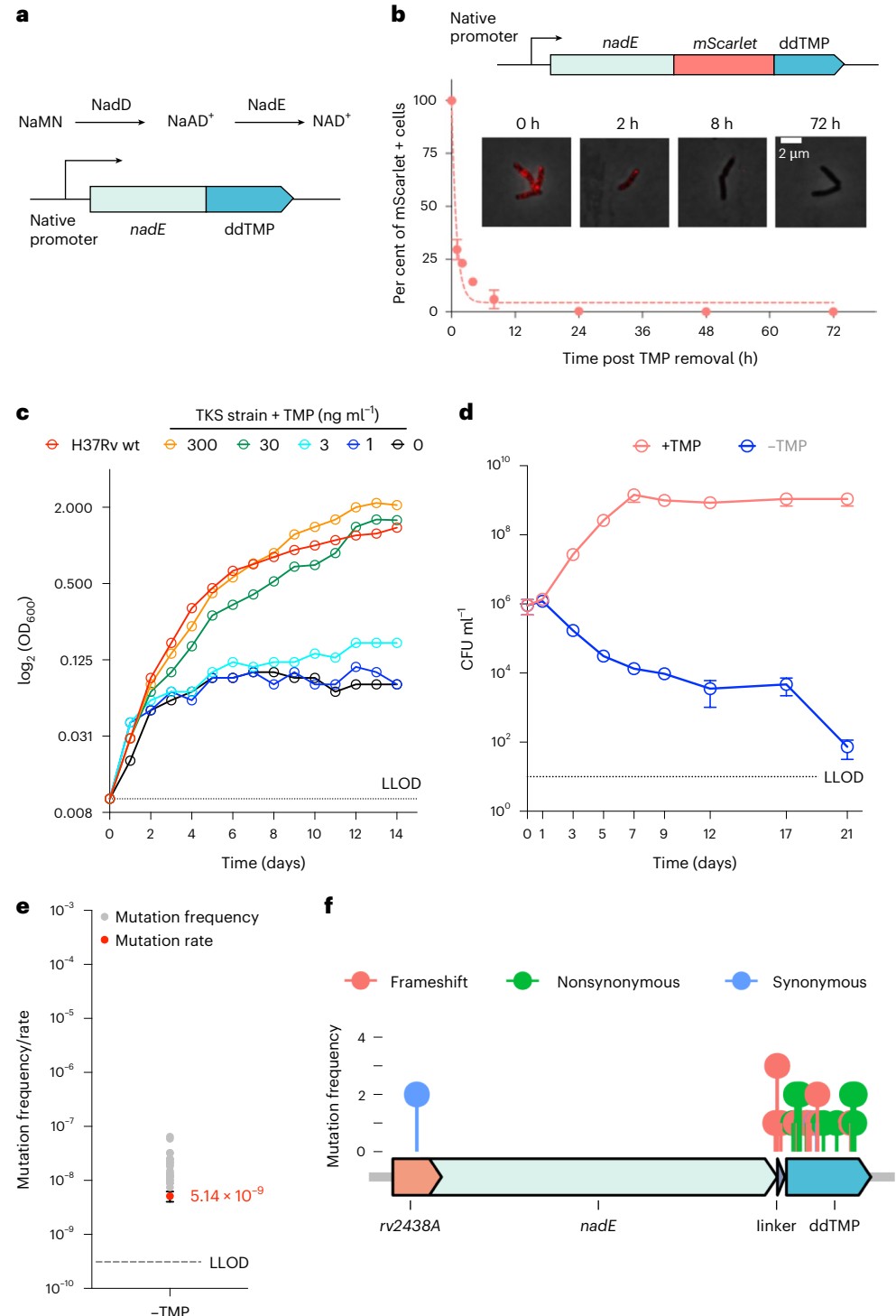

**Fig. 3 | TMP-dependent NadE degradation leads to Mtb death. a**, The genetic construct of tagging Mtb gene *nadE* with C-terminus ddTMP. **b**, The degradation kinetics of NadE-mScarlet-ddTMP were measured by quantifying mScarlet intensity per cell via microscope. The degradation curve is fitted into a one-phase exponential decay model. **c**, The growth curve of the Mtb *nadE-ddTMP* strain with various TMP concentrations in 7H9 media. Each dot represents the mean of biological triplicates. The data are presented as mean ± standard deviation.

$OD_{600}$, optical density at 600 nm. **d**, The kill curve of the Mtb *nadE-ddTMP* strain in the absence of TMP. Each dot represents the mean of biological triplicates. The data are presented as mean ± standard deviation. **e,f**, The mutation rate of Mtb *nadE-ddTMP* strain (**e**) and its escape mutation density plot (**f**) are determined by whole-genome sequencing. The escape rate is determined via escape frequency of 20 independent cultures. The escape rate is presented as the mean, with error bars showing standard deviation. The dashed line indicates the LLOD.

to escape. Through fluctuation assays, we determined the escape rate of the NadE-ddTMP kill switch to be ~$5.1 \times 10^{-9}$ per genome per generation (Fig. 3e). Subsequent whole-genome sequencing of 20 escape mutants of NadE-ddTMP revealed that most resistance

mutations were frameshift or nonsynonymous mutations at the *nadE* C-terminus or within the ddTMP degron region (Fig. 3f and Supplementary Table 2). This pattern indicated that escape probably occurred due to the functional ablation of the ddTMP degron. We

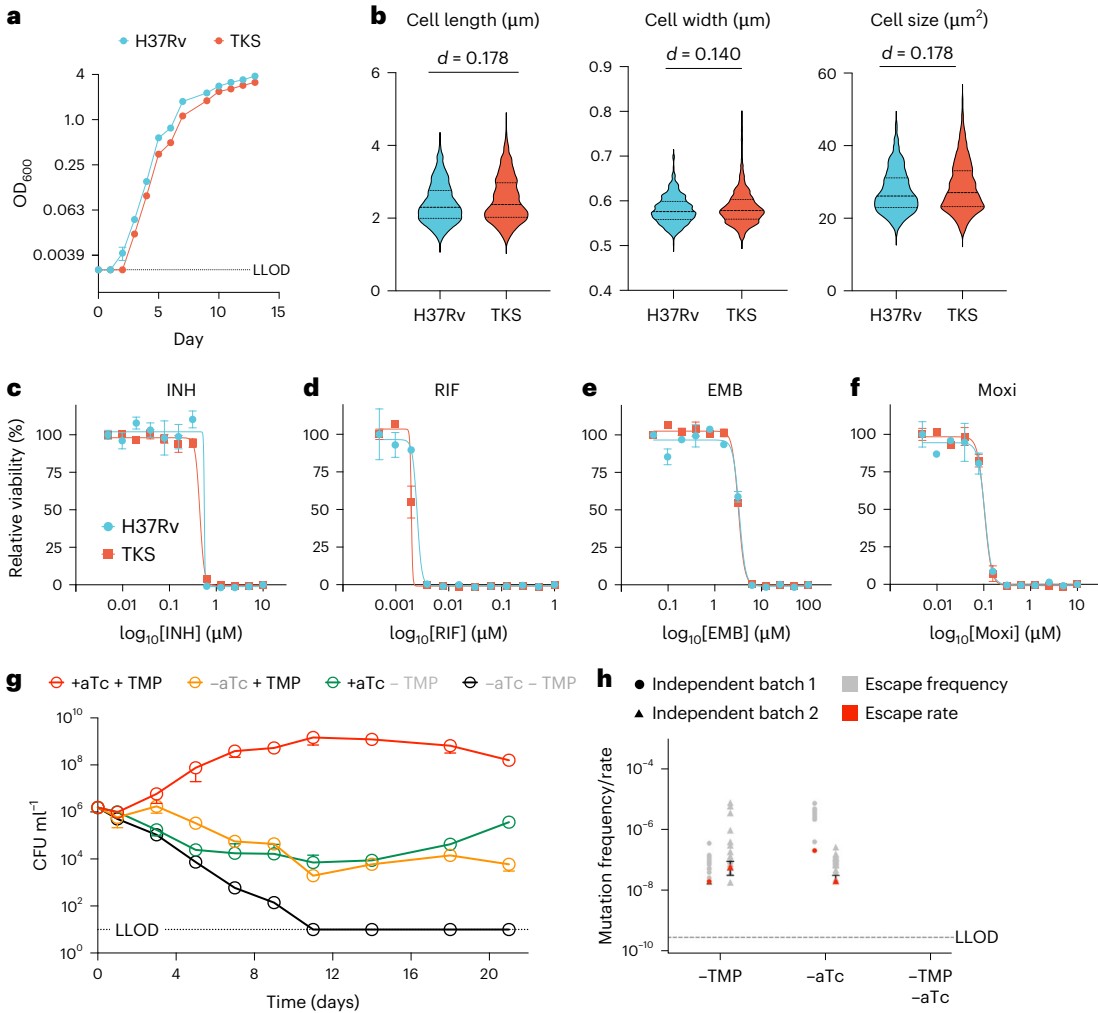

**Fig. 4 | The TKS strain manifests normal fitness at permission condition yet rapid killing and low escape rate at restrictive condition. a**, The growth curve of TKS and H37Rv strains at permissive condition. The dotted line shows the LLOD as OD 0.001. Each dot represents the mean of biological triplicates. The data are presented as mean ± standard deviation. **b**, The quantified bacterial morphology of strains H37Rv (*n* = 1,068) and TKS (*n* = 1,614). Cohen's *d* was calculated to show effect size between H37Rv and TKS strains. **c**–**f**, The MIC of H37Rv and TKS strains against isoniazid (INH) (**c**), rifampin (RIF) (**d**), ethambutol (EMB) (**e**) and moxifloxacin (Moxi) (**f**) at permissive condition. Each dot represents the mean of biological triplicates. The curve shows a sigmoidal regression of relative viability as a function of antibiotics concentration on a logarithmic scale. **g**, The in vitro kill curve of TKS strain in the absence of aTc and/or TMP. The dotted line shows the LLOD as 10 CFU ml⁻¹. Each dot represents the mean of biological triplicates. The data are presented as mean ± standard deviation. **h**, The escape rate of the TKS strain in the absence of aTc and/or TMP from two independent experiments. Each experiment contains 20 independent bacteria culture for escape frequency quantitation. In each experiment, the escape rate is calculated from escape frequency of 20 culture and is presented as the mean, with the error bars showing standard deviation. The dashed line shows the LLOD as 3.8 × 10⁻¹⁰. The error bars represent standard deviation.

also identified a synonymous mutation in the *nadE* upstream region, which probably reflects a promoter mutation inducing higher *nadE* expression to counteract proteolysis stress (Fig. 3f and Supplementary Table 2). These escape mutations indicate that the Mtb killing effect was indeed attributed to NadE degradation, devoid of off-target consequences.

## TKS induction in Mtb is bactericidal in vitro

A successful Mtb human challenge strain necessitates growth kinetics resembling the wild-type strain under permissive conditions, while ensuring rapid, low-escape-rate killing under restrictive conditions. To achieve this, we combined the two lysin switches with NadE-ddTMP, yielding a TKS Mtb H37Rv strain. The TKS strain's growth kinetics closely mirrored those of wild-type H37Rv when cultivated under permissive conditions with aTc and TMP supplementation (Fig. 4a). Under permissive conditions, disparities with small effect size (Cohen's *d* < 0.200) in cellular morphology were observed between the TKS and H37Rv strains

(Fig. 4b). Furthermore, susceptibility to first-line and second-line antimycobacterial agents, such as isoniazid, rifampin, ethambutol and moxifloxacin, remained indistinguishable between the TKS strain and H37Rv (Fig. 4c–f). However, in the absence of aTc and TMP, the TKS strain exhibited rapid killing as reflected by an ~3 log decline in CFU per week. In axenic culture, complete sterilization was achieved within 11 days after aTc and TMP removal (Fig. 4g). Depleting either TMP or aTc individually did not yield such rapid and complete sterilization, implying a synergistic bacterial-killing effect between the phage-lysin and ddTMP degron kill switches.

The design of the TKS strain enabled us to measure the escape rates of each system independently. We determined escape rates with two independent batches of bacteria culture using fluctuation analysis. In the presence of aTC, we found the escape rate for the ddTMP degron ranging from 2.0 × 10⁻⁸ to 6.0 × 10⁻⁸ per genome per generation. In the presence of TMP, we found the dual-lysin escape rate ranging from 2.1 × 10⁻⁸ to 2.1 × 10⁻⁷ per genome per generation (Fig. 4h).

Whole-genome sequencing indicated that escape mutations in the ddTMP degron system resulted from the truncation of the ddTMP degron tag, while escape mutations associated with the dual-lysin kill switches arose from disruptions and deletions within the lysin operons (Supplementary Table 3). Importantly, the escape rate fell below the lower limit of detection within our current experimental setup under the stringent restrictive conditions, encompassing depletion of both aTc and TMP ($<3.8 \times 10^{-10}$ per genome per generation). Since escape mutation mechanisms are independent, the theoretical escape rate should be $<10^{-15}$.

## TKS Mtb is cleared in mice without relapse

To test the TKS strain virulence in permissive conditions and its clearance in restrictive conditions, we infected C57BL/6 mice with the TKS strain via the aerosol route. PK/PD studies on doxy and TMP suggested that dietary supply or removal of doxy and TMP promptly modulated their levels in mouse tissues and plasma, either above or below the threshold necessary for the TKS strain survival[26] (Supplementary Table 4). Therefore, we fed mice chow supplemented with TMP and doxy for 28 days and then switched to regular chow until day 140 (Fig. 5a). The TKS strain established infection under permissive conditions, resulting in pulmonary CFU ranging from $10^3$ to $10^4$ by day 28. The pulmonary burden under permissive conditions reached a plateau of $10^5$ CFU (Fig. 5b). Dissemination to the spleen was observed at ~day 28, as evident from two out of five mice with a spleen bacterial burden of ~100 CFU. The spleen CFU stabilized at approximately $10^3$–$10^4$ CFU after day 56 (Fig. 5c). Upon discontinuing TMP and doxy supplementation at day 28, the pulmonary bacterial burden fell at a rate of ~0.75 log CFU reduction per week. Extrapolation suggested complete pulmonary clearance by day 96, with no detectable live bacteria at days 112 and 140 under restrictive conditions (Fig. 5b). Only one out of five mice had detectable bacteria in the spleen on days 56 and 84, and all mice cleared splenic infection on days 112 and 140 (Fig. 5c).

To investigate immune responses, we examined pulmonary CD4$^+$ and CD8$^+$ T cell profiles on day 84 in mice with or without doxy/TMP supplementation. TKS infection under permissive conditions prompted infiltration of activated T cells (CD11a$^{high}$, CD69$^+$, KLRG1$^+$ and PD-1$^+$) (Fig. 5d,f and Extended Data Fig. 1), along with the recruitment of effector and resident memory T cells (Fig. 5e,g and Extended Data Fig. 1). Within the T cell population expressing the above markers, we observed substantially attenuated CD4 and CD8 responses under restrictive conditions compared with permissive conditions. After the TKS strain clearance at d84, the CD4 response is indistinguishable from uninfected controls. Similarly, after the TKS strain clearance, the CD8 response is largely similar to uninfected control at day 84, except for a slightly increased proportion of CD8$^+$ effector T cells among the total CD8$^+$ T cell population.

To assess the potential for relapse under stringent conditions, we utilized the immunocompromised SCID and Rag$^{-/-}$ mouse models (Fig. 5h,j). We administered TMP and doxy supplemented chow from days 1 to 14 in both mouse models, switched to chow without supplements, then switched back to supplemented chow from days 98 to 168 in the SCID model and days 56 to 126 in the Rag$^{-/-}$ model. The TKS strain established infection with $5 \times 10^3$ pulmonary CFU in SCID mice and $2.5 \times 10^3$ CFU in Rag$^{-/-}$ mice by day 14. Upon depleting TMP and doxy, bacteria cleared at a rate of ~1.5 log CFU per week in the SCID model and 0.9 log CFU per week in the Rag$^{-/-}$ model. There was no detectable bacterial growth from lung homogenates at days 56 and 98 in the SCID model and at day 56 in the Rag$^{-/-}$ model. After return to doxy and TMP supplementation for 10 weeks, no viable bacteria were detected at day 168 in the SCID model and day 126 in the Rag$^{-/-}$ model in the lungs (Fig. 5i,k). In addition, no viable bacteria were found in other organs including spleen, mediastinal lymph node, liver and bone marrow (Extended Data Figs. 2 and 3). Thus, the TKS strain did not reemerge under permissive conditions, even in the absence of adaptive immunity, once it had been eliminated by kill-switch induction.

We observed tight regulation of TKS strain growth and death through doxy/TMP supplementation, with discontinuation leading to rapid clearance and attenuation of the immunopathological response. Importantly, no relapse occurred even in the absence of an intact immune system.

## Discussion

The burgeoning interest in CHIM studies, aimed at expediting TB vaccine development, has emphasized the critical need for an engineered Mtb challenge strain to effectively address safety concerns. When assessing vaccine efficacy, the challenge strain must demonstrate consistent replication to ensure robust immunogenicity, while also enabling rapid killing at the conclusion of the study. This rationale led us to design a challenge strain reliant on exogenous molecules, triggering bacterial killing upon the removal of small molecule supplementation. To meet these goals, we have engineered an Mtb strain equipped with three distinct kill switches. This engineered strain exhibits growth kinetics and antibiotic susceptibility akin to the wild-type Mtb in the presence of doxy and TMP. In the absence of doxy and TMP, it is rapidly eradicated in vivo.

Safety is the foremost concern for the TB human challenge strain. The most noteworthy attribute of the TKS strain is its robust killing in the absence of doxy and TMP supplementation. The removal of doxy prompts the expression of lysins L5 and D29, culminating in cell wall damage and bacterial lysis. The absence of TMP triggers ClpC1-mediated degradation of NadE, a crucial metabolic enzyme in the final step of NAD$^+$ biosynthesis. The amalgamation of these three orthogonal killing mechanisms yields a low escape rate and rapid elimination kinetics. In the TKS strain, the escape rates are from ~$10^{-8}$ to $10^{-7}$ per genome per generation for both the ddTMP degron kill switch and dual-lysin kill switch. Given the independent mechanisms of two kill switches, the $10^{-15}$ theoretical combined escape rate is calculated as a product of escape rate of each individual kill switch. We set a rigorous goal of an escape rate of ≤$10^{-12}$ due to practical reasons for experiment validation. It takes 10 litres of Mtb culture to measure an escape rate at a magnitude of $10^{-12}$ per genome per generation via fluctuation test. That

**Fig. 5 | The TKS strain is rapidly cleared in mouse without discernible relapse. a**, The scheme of clearance study in C57BL/6 mice. The mice were kept on doxy/TMP chow 3 days before infection until day 28 post infection. Lung and spleen CFU were enumerated at indicated timepoint. **b,c**, Lung (**b**) and spleen (**c**) CFU of the C57BL/6 mouse experiment. Each dot represents the CFU mean from $n = 5$ mice. The error bars denote standard deviations. The $P$ values were calculated via two-tailed Student's $t$-test compared between CFU in regular chow group and the doxy/TMP chow group at a given timepoint. **d**, The proportion of CD4$^+$ T cells of the C57BL/6 mouse with activation markers at day 84. **e**, The proportion of CD4$^+$ memory T cells of the C57BL/6 mouse at day 84. **f**, The proportion of CD8$^+$ T cells of the C57BL/6 mouse with activation markers at day 84. **g**, The proportion of CD8$^+$ memory T cells at day 84. The $P$ values in **d**–**g** were calculated by a Student's $t$-test with a multiple comparison corrected with Tukey method. The grey data are from uninfected mice, blue data are from mice switched to regular chow and green data are from mice with doxy/TMP chow. **h,j**, The scheme of relapse study in SCID mice (**h**) and Rag$^{-/-}$ mice (**j**). The mice were kept on doxy/TMP chow 3 days before infection until day 14 post infection. The mice were then fed with regular chow until indicated timepoint. The doxy/TMP chow were later given to mice for 10 weeks for relapse assessment. Lung CFUs were enumerated at indicated timepoints. **i,k**, The kinetics of lung CFU enumerated in the SCID (**i**) and Rag$^{-/-}$ (**k**) relapse experiment. The bacteria burden was assessed with $n = 5$ mice at each timepoint. Each dot represents the CFU from one mouse. The error bars denote the mean and standard deviation. The dotted line shows the CFU LLOD. Panels **a**, **h** and **j** created with BioRender.com, released under a Creative Commons Attribution-NonCommercial-NoDerivs 4.0 International license.

high volume is at the limit of our BL3 safety capacity. The calculated rate derived from experimental measurements exceeds that goal. We reasoned that it is unlikely for the TKS strain to develop stepwise escape mutations in each kill switch sequentially due to an uneven tissue distribution of doxy and TMP. Previous PK/PD studies on doxy and TMP indicate good oral absorbance, high serum concentrations and excellent tissue penetration[26–29]. The pulmonary levels of both

drugs are above the minimum concentration required for growth of the TKS strain.

A second safety goal is to attain rapid clearance of bacteria after withdrawal of antibiotics. We saw rapid clearance rates in mice regardless of their immune status. Importantly, the TKS strain did not cause relapse disease in immunocompromised SCID and Rag⁻/⁻ mice, even after 10 weeks of TMP and doxy reintroduction. Collectively, these

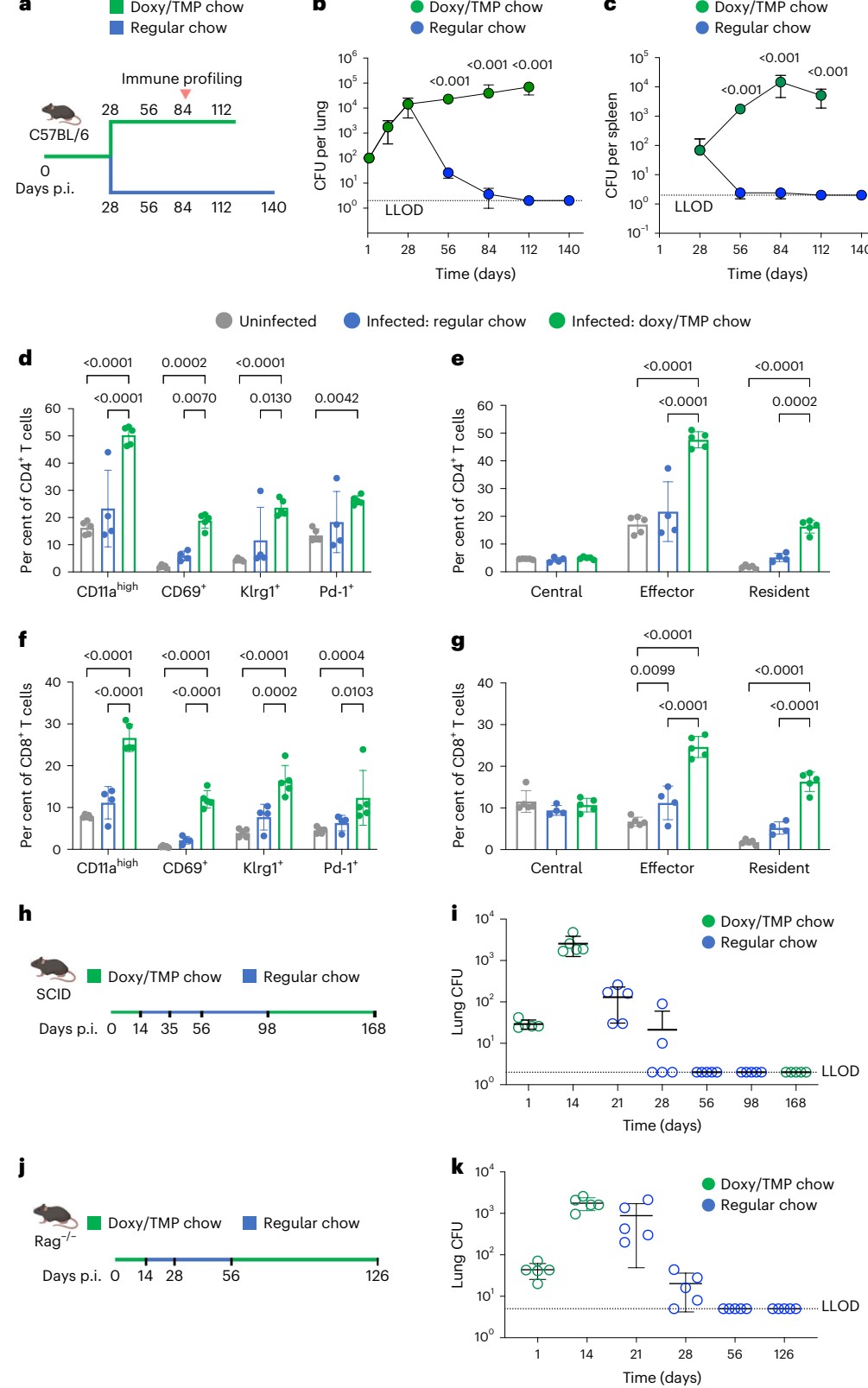

growth and killing attributes derived from the kill switches suggest that this strain could be safely used for TB human challenge.

Thus far, no animal model reliably predicts vaccine efficacy in humans. Although mice are widely used in TB immunology studies, they fail to replicate human TB disease due to cross-species differences such as distinct macrophage responses, the absence of pulmonary granuloma structures and differing antigen presentation repertoires. These disparities impede TB vaccine development grounded solely in mouse studies. NHP studies model human TB better than mice, though, again, we do not yet know their predictive power. A human challenge model may allow more rapid identification of human immune correlates in TB infection and protection, facilitating vaccine efficacy testing and validating correlates identified in previous NHP and mouse studies. A recent human challenge study in the UK, involving the bronchial inoculation of BCG or PPD in humans, revealed novel insights distinct from prior in vitro studies[16]. However, a limitation to the use of BCG as a CHIM strain is the lack of expression of certain antigens, particularly those encoded in the RD1 region, some of which are included in vaccine platforms. Thus, an Mtb challenge strain would be more appropriate in assessing vaccine effectiveness. Future TB CHIM work using the attenuated Mtb TKS strain could compare the immunogenicity between the TKS strain and wild-type Mtb and illuminate the host–pathogen interaction more relevant to Mtb infection than the BCG and PPD model.

Our study has limitations. First, in animal experiments, bacteria grew and cleared in a manner similar to what we observed in vitro. However, the kinetics of growth and clearance could well differ in other species and these results certainly do not perfectly predict what would occur in human infection. Assessing kill kinetics and relapse rates in NHPs will help validate safety for human studies. It will also be important to determine a 'safe' bacterial burden that minimizes the risk of pathologic injury to participants. Second, alternative challenge routes, apart from aerosol, could be explored for the TKS strain to assess its ability to detect a vaccinal effect. Considering the distinct tissue pharmacokinetics and pharmacodynamics of doxy and TMP, it will be important to benchmark growth and killing kinetics in multiple tissues for the TKS strain. Third, a sensitive and non-invasive detection method will be important to track strain clearance in vivo. Ideally, the detection approach should be non-invasive to minimize costs and efforts for clinical cohorts and volunteers.

In conclusion, we have successfully engineered an Mtb TKS strain enabling mouse infection comparable with the wild-type strain under permissive conditions and displaying rapid and complete sterilization without relapse under restrictive conditions. We propose the TKS strain as a suitable challenge strain for a TB CHIM, advancing TB vaccine efficacy studies and shedding light on human immune responses to Mtb infection.

## Methods

### Animal ethics statement
The mice were housed under a 12 h light–dark cycle, ambient temperature and humidity condition in BSL3 vivarium at Weill Cornell Medicine and Harvard T.H. Chan School of Public Health. Animal care and experimental procedures were conducted with the approval of the Institutional Animal Care and Use Committees (IACUC) of Harvard Medical School, Harvard T.H. Chan School of Public Health and Weill Cornell Medicine.

For NHP studies, all experimental manipulations, procedures, protocols and care of the animals were approved by the University of Pittsburgh School of Medicine IACUC. The protocol assurance number for our IACUC is A3187-01. The specific protocol approval number is 18124087. The University of Pittsburgh's IACUC adheres to national guidelines established in the Animal Welfare Act (7 U.S.C. Sections 2131–2159) and the Guide for the Care and Use of Laboratory Animals (eighth edition), as mandated by the US Public Health Service Policy.

### Strains, media and culture conditions
All Mtb strains are H37Rv derivatives, except the dual-lysin strain used for macaque studies, which was constructed in the Erdman background. All Msm strains are mc$^2$155 derivatives. The wild-type strains H37Rv or mc$^2$155 were grown in Middlebrook 7H9 broth or 7H10 agar supplemented with 0.5% glycerol, 0.05% Tween-80 and 1× oleic acid–albumin–dextrose–catalase (OADC, Middlebrook 212351) supplement (Mtb) or 1× albumin–dextrose–catalase supplement (Msm). For the permissive condition of the dual-lysin strain, the media was supplemented with aTc (0.5–1.0 µg ml$^{-1}$), kanamycin (20 µg ml$^{-1}$) and zeocin (25 µg ml$^{-1}$). For the permissive condition of the NadE-ddTMP strain, the medium was supplemented with TMP (50 µg ml$^{-1}$) and hygromycin (50 µg ml$^{-1}$). For the TKS strain, the medium was supplemented with aTc (0.5–1.0 µg ml$^{-1}$), TMP (50 µg ml$^{-1}$), kanamycin (20 µg ml$^{-1}$), hygromycin (50 µg ml$^{-1}$) and zeocin (25 µg ml$^{-1}$).

### Generation of strains
To create the dual-lysin kill-switch Mtb strain, H37Rv or Erdman was transformed with plasmids pGMCK3-TSC10M-TsynE-pipR-SDn-PptR-D29L and pGMCgZni-TSC38S38-P749-10C-L5L, incorporating the D29-lysin and L5-lysin into Mtb L5 and Giles sites, respectively. aTc was supplemented at 0.5–1.0 µg ml$^{-1}$.

To create the NadE-ddTMP Mtb strain, H37Rv with plasmid pKM461 for ORBIT recombineering[30] was transformed with an oligo (Supplementary Table 4) and a payload plasmid pJW461. The oligo contains an *attP* site flanked by 70 nucleotides at 5′- and 3′-end, respectively, homologous to the insertion site. The pKM461 contains an *attB* site and ddTMP sequence. These constructs resulted in labelling the ddTMP tag at the C-terminus of NadE mediated via ORBIT as described previously[30]. The NadE-ddTMP strain was maintained with TMP supplemented at 50 µg ml$^{-1}$.

To create the TKS strain, the NadE-ddTMP strain was transformed with plasmids pGMCK3-TSC10M-TsynE-pipR-SDn-PptR-D29L and pGMCgZni-TSC38S38-P749-10C-L5L to incorporate the D29-lysin and L5-lysin into the Mtb genome with aTc supplemented at 0.5 µg ml$^{-1}$.

### Fluctuation analysis
The fluctuation analysis was performed as previously reported[31]. In brief, the Mtb strain was inoculated in permissive conditions (0.5–1.0 µg ml$^{-1}$ aTc and 50 µg ml$^{-1}$ TMP) in 7H9 media supplemented with OADC in the presence of antibiotics (20 µg ml$^{-1}$ kanamycin, 25 µg ml$^{-1}$ zeocin and 50 µg ml$^{-1}$ hygromycin). After reaching an optical density (OD) of 1.0, the culture was diluted to $n = 20$ 4 ml aliquots with 10,000 bacteria. The diluted culture was grown for 11–14 days in 7H9 + OADC media in permissive conditions in the presence of antibiotics. Once the OD was at 1.0, bacteria were washed for three times and resuspended in 400 µl 7H9 + OADC without aTc or TMP. Four aliquots of bacteria were streaked onto 7H10 + OADC plates supplemented with 0.5–1.0 µg ml$^{-1}$ aTc and 50 µg ml$^{-1}$ TMP for bacteria count, and the rest aliquots were spread onto 7H10 + OADC plates with either aTc or TMP or neither supplement. According to the Ma, Sarkar and Sandri method[32,33], the estimated number of mutations per culture ($m$) was inferred by number of mutant ($r$) colonies observed on plates. The escape rate was calculated by dividing $m$ by $N_t$ (the number of cells plated for each culture). The Mann–Whitney $U$ test was used to statistically compare escape rates between two groups. The lowest detection limit was calculated on the basis of an assumption that only one colony could be observed in all 20 independent cultures.

### Western blots
The western blots follow protocol from a previous study[34]. The mouse IgG2b monoclonal RpoB antibody (no. 8RB13, Cell Signaling) and anti-myc antibody (no. 71D40, Cell Signaling) were diluted 1:1,000. The IRDye 680RD goat IgG (H+L) anti-mouse and IRDye 800CW goat IgG (H+L) anti-rabbit (LI-COR) fluorescent secondary antibodies were diluted 1:15,000.

## Mtb flow cytometry

Mtb cells were fixed in 2% paraformaldehyde overnight and removed from the BSL3 facility. The fixed bacilli were quenched with 200 mM Tris–HCl (pH 7.5) for 5 min at room temperature and resuspended in 1× phosphate-buffered saline (PBS) buffer with 0.1% Triton X-100. To suppress signals from noise or cell debris, two event triggers (thresholds) on forward scatter (peak height >1.5 and side scatter area >1.0) were used upon recording. To remove cellular aggregation, stringent gate settings were manually defined via FlowJo v.10.8 to exclude events with strongly correlated forward scatter area and side scatter area measures (large and compact particles), as well as events with disproportional forward scatter area and forward scatter peak height measures (morphological outliers). After event filtration, the $\log_{10}$-transformed red fluorescence intensity peak height (denoted TdTomato-A) was used to represent the abundance of intracellular red fluorescence protein.

## Microscope imaging and analysis

Mtb cultures were fixed with 2% paraformaldehyde for 1 h. Paraformaldehyde quenching and cell disaggregation were achieved by washing the fixed bacilli with a customized solution containing 200 mM Tris–HCl (pH 7.5), 1% (w/v) Triton X-100, 0.67% (v/v) xylenes and 0.33% (v/v) heptane[35]. The cells were then washed once with PBS–Triton (0.1% v/v) and seeded onto moulded 1.8% agarose in PBS for imaging. Phase contrast and fluorescence micrographs were collected via a Plan Apo 100× 1.45 NA objective using a Nikon Ti-E inverted, widefield microscope equipped with a Nikon Perfect Focus system with a Piezo Z drive motor, Andor Zyla sCMOS camera and NIS Elements (v4.5). mScarlet signal was acquired using a six-channel Spectra X light-emitting diode light source and the Sedat Quad filter set. The excitation and emission filters were excitation 550/15 nm and emission 595/25 nm, respectively. Semiautomated imaging was carried out using a customized Nikon JOBS script to locate imaging fields of interest, 9–12 images were taken for each sample. Cell segmentation and quantification was performed using our previously published Python pipeline, MOMIA[36]. We applied R function lsr::cohensD() to calculate effect size between H37Rv and TKS strains.

## Kill curve assay

Mid-log phase Mtb cultures (OD of 0.3, $10^8$ CFU ml$^{-1}$) grown in permissive conditions were washed three times with 7H9 + OADC without aTc or TMP, followed by dilution to $10^6$ CFU ml$^{-1}$ (OD of 0.003). The diluted cultures were supplemented with aTc or TMP or both, and bacterial CFU was enumerated by plating serial dilutions on 7H10 + OADC plates at multiple timepoints. The experiment was performed in biological triplicates.

## Mtb genomic DNA extraction

The Mtb culture at OD of 1.0 was pelleted in buffer TE-NaCl followed by delipidation with chloroform:methanol (2:1). The delipidated bacteria were reconstituted in TE-NaCl followed by lysozyme (100 µg ml$^{-1}$) digestion at 37 °C overnight and proteinase K (100 µg ml$^{-1}$) digestion in 0.01% SDS at 50 °C for 2 h. The digested lysate was treated with equal volume of phenol:choloroform (1:1, pH 8) followed by centrifugation, and the aqueous phase was treated with one-half volume of chloroform. After centrifugation, the genome DNA (gDNA) in the aqueous phase was precipitated with a one-tenth volume of 3 M sodium acetate (pH 5.2) and one volume of isopropanol at −20 °C overnight. The gDNA pellet was washed with 70% ethanol twice and dissolved in nuclease-free water (Invitrogen). The gDNA concentration was measured by Qubit fluorometer.

## Mtb whole-genome sequencing and SNP calling

The gDNA library was prepared with Nextera XT DNA Library Preparation Kit (Illumina, FC-131-1096) and Nextera XT Index Kit v2 (Illumina, FC-131-2001). The pooled library was sequenced on MiSeq platform (Illumina) with paired-end strategy. The reads were trimmed by sickle (version 1.33)[37] to preserve reads with a Phred base above 20 and read length longer than 30, followed by mapping against the H37Rv reference genome (ASM19595v2) using bwa[38]. The duplicated reads were then removed by SAMtools[39]. We used VarScan (version 2.3.9)[40] to call single nucleotide polymorphism (SNP) variants with the parameters --min-coverage 3--min-reads2 2--min-avg-qual 20--min-var-freq 0.01--min-freq-for-hom 0.9--p-value 99e-02--strand-filter 0.

## MIC

The minimal inhibitory concentration (MIC) values of drugs were determined following the microplate-based Alamar Blue assay method as previously described[41]. Twofold serial dilutions of drug were prepared in sterile polystyrene 96-well round-bottom plates (CLS3795, Corning) with Middlebrook 7H9 + OADC media, 100 µl per well. Mtb strains were 1:1,000 diluted in 7H9 + OADC medium at mid-logarithmic stage of growth (OD of 0.4). A total of 50 µl of diluted bacterial suspensions were inoculated into each well, followed by incubation at 37 °C for 7 days. A total of 20 µl alamarBlue reagent (Invitrogen, Frederick), freshly mixed with 12.5 µl 20% Tween-80, was added into each well, followed by 24 h incubation at 37 °C. The absorbance was read at 570 nm, with reference wavelength 600 nm, using a microplate reader. The MIC endpoint was defined as the lowest concentration of the test agent that produced at least 90% reduction in absorbance compared with that of the drug-free control.

**Mouse experiments clearance study.** Female C57BL/6J mice (The Jackson Laboratory, 000664) at 6–8 weeks old were randomly assigned to experiment groups. The mice were aerosol infected with the TKS strain with 50–100 CFU. We used chow supplemented with 2,000 ppm doxy and 1,600 ppm TMP (Research Diets) as permissive chow. The mice were fed with permissive chow starting 3 days before infection through 28 days post infection. After day 28, the mice on in the permissive arm were fed with permissive chow through day 112, whereas mice in the restrictive arm were fed with regular chow through day 140. The bacterial burden was determined by plating lung and spleen homogenates at day 1, 28, 56, 84, 112 and 140 post infection from $n = 5$ mice. No statistical methods were used to predetermine sample sizes, but our sample sizes are similar to those reported in previous publications[42,43]. The data distribution was assumed to be normal but this was not formally tested.

**Lung cell isolation.** Murine right lung lobes were collected into 2 ml cold PBS in 5 ml conical tubes kept in a cooling rack. Once all lungs were collected, the right lung lobes were transferred into gentleMACS C tubes (Miltenyi Biotec) containing 2.5 ml prewarmed digestion buffer consisting of filter-sterilized PBS (with calcium and magnesium) 0.5% bovine serum albumin with freshly added DNase I (Roche, 10104159001) at 100 µg ml$^{-1}$ and Collagenase IV (Worthington, LS004186) at 150 units per millilitre. Lungs were dissociated in a gentleMACS Dissociator (Miltenyi Biotec), followed by incubation at 37 °C for 45 min. The digested tissue was dissociated again and passed through a 70 µm cell strainer into a 50 ml conical tube. The strainer was rinsed with 2 ml of PBS 0.5% bovine serum albumin. The red blood cells were lysed with eBioscience 1× RBC Lysis Buffer (Thermo Scientific 00-4333-57). The lung cells were washed in PBS, resuspended in RPMI complete medium (RPMI-1640 supplemented with 10% FBS, 2 mM GlutaMax, 10 mM HEPES buffer, 0.05 mM 2-mercaptoethanol and 1% penicillin–streptomycin) and enumerated using the automated cell counter Countess II (Thermo Fisher Scientific).

**Flow cytometry of lung cells.** Lung cell suspensions were washed with PBS twice, stained for viability with the live/dead dye Zombie UV (BioLegend, 423107) for 10 min at 4 °C, washed with Cell Staining Buffer (BioLegend 420201) and incubated with Fc block (purified

anti-mouse CD16/CD32 antibody, BioLegend 101302) at 1:200 in Cell Staining Buffer for 10 min at 4 °C. After one wash in Cell Staining Buffer, the cells were incubated with fluorochrome-conjugated monoclonal antibodies (mAbs) diluted at 1:200 into a 1:3 solution of Brilliant Staining Buffer (BD 563794): Cell Staining Buffer, for 45 min at 4 °C. The cells were then washed twice in Cell Staining Buffer and fixed with Fixation Buffer (BioLegend 420801) for 30 min at 4 °C. In all incubation steps cells were protected from light. Fluorescence minus one controls were stained alongside samples. Fluorochrome-conjugated mAbs used were anti-mouse CD69-BB700 (H1.2F3, BD Biosciences), CD44-FITC (IM7, BioLegend), CD62L-BV650 (MEL14, BD Biosciences), CD11a-BV605 (2D7, BD Biosciences), KLRG1-PE-Cy7 (2F1, BioLegend), PD-1-BV421 (29F.1A12, BioLegend), CD8-BUV496 (53-6.7, BD Biosciences), CD3-BUV395 (17A2, BD Biosciences), CD4-APC-H7 (GK1.5, BD Biosciences) and CD45-APC (30-F11, BioLegend). The data were acquired with a BD FACSymphony A5 Cell Analyzer in the Flow Cytometry Core at Weill Cornell Medicine and analysed with FlowJo v10.8 software (BD Life Sciences).

**Relapse study SCID mouse model.** Female B6.Cg-*Prkdc*[scid]/SzJ mice (The Jackson Laboratory, 001913) at 6–8 weeks old were randomly assigned to experiment groups. The mice were aerosol infected with the TKS strain with 50–100 CFU. The mice were fed with permissive chow (2,000 ppm doxy and 275 ppm TMP) starting 3 days before infection and through 14 days post infection. After day 14, the mice were switched to regular chow until day 98 to clear infection. The mice were then back on permissive chow for 10 weeks until day 168 to assess relapse. The bacteria burden was assessed at day 1, 14, 21, 28, 56, 98 and 168 from lung and spleen homogenates (*n* = 5). No statistical methods were used to predetermine sample sizes, but our sample sizes are similar to those reported in previous publications[42,43]. The data distribution was assumed to be normal but this was not formally tested. For the Rag[−/−] mouse model, female B6.129S7-*Rag1*[tm1Mom]/J mice (The Jackson Laboratory, 002216) at 6–8 weeks old were randomly assigned to experiment groups. The mice were aerosol infected with the TKS strain with 50–100 CFU. The mice were fed with permissive chow (2,000 ppm doxy and 275 ppm TMP) starting 3 days before infection and through 14 days post infection. After day 14, mice were switched to regular chow until day 56 to clear infection. The mice were then back on permissive chow for 10 weeks until day 126 to assess relapse. The bacteria burden was assessed at day 1, 14, 21, 28, 56 and 126 from multiple organs, including lung, spleen, mediastinal lymph node, liver and bone marrow (*n* = 5). The bone marrow cells were flushed from both femurs and tibiae with 4 ml lysis buffer (PBS + aTc + TMP) and passed through a 70 μm nylon mesh cell strainer. No statistical methods were used to predetermine sample sizes, but our sample sizes are similar to those reported in previous publications[42,43]. The data distribution was assumed to be normal but this was not formally tested.

**Macaque studies**

Mauritian cynomolgus macaques (*Macaca fascicularis*) were obtained from Bioculture Mauritius (*n* = 6, all males, 6–9 years old). The macaques were challenged with 6 CFU Mtb Erdman dual-lysin strain via bronchoscope. Doxy was administered daily in food treats at 40 mg kg⁻¹ (orally once daily) beginning 1 day before Mtb infection. Doxy was discontinued after 2 weeks for four macaques, while the other two macaques were administered doxy for a total of 6 weeks. Comprehensive necropsies were performed at 6 weeks post challenge. The gross pathology scores reflect the presence of any granulomas or other pathologies, size of lymph nodes and any other TB-related disease. A score of 7 is considered normal/uninfected. The number of involved lymph nodes reflects thoracic lymph nodes with visible granulomas or that are CFU positive. Any TB-related pathologies (such as granulomas), all lung lobes, all thoracic and peripheral lymph nodes, spleen and liver were collected and individually processed into single-cell suspensions.

The samples were plated on aTc containing 7H11 plates, incubated at 37 °C with 5% $CO_2$ and counted at 3 and 6 weeks post plating.

**Reporting summary**

Further information on research design is available in the Nature Portfolio Reporting Summary linked to this article.

## Data availability

Escape mutant whole-genome sequences of the TKS Mtb strains have been deposited to the BioProject database under accession code PRJNA1194322. The numeric source data have been provided for Figs. 1c,f, 3b–e, 4a,c–h and 5b–g,i,k. The uncropped western blot gel images have been provided as source data for Fig. 2a–d. Source data are provided with this paper.

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

## Acknowledgements

We thank M. Chao for discussions during this work. We thank R. A. Olvera, H. Kim and L. Pipkin for technical help with mouse infections. We appreciate the dedication and expertise of the veterinary and research technicians and computed tomography combined with positron emission tomography (PET–CT) imaging personnel at the University of Pittsburgh School of Medicine. This work was supported by NIH R01 AI135629 and Bill and Melinda Gates Foundation INV-009003 and OPP1135516.

## Author contributions

E.J.R., D.S., S.M.F and S.E. conceptualized and designed the study. H.S., J.B.W., T.K., K.L., Y.L., S.E. and D.S. designed and constructed the dual-lysin Mtb kill-switch strain, performed the mutation rate experiment and profiled kill curve in axenic culture and in C57BL/6 mice. M.R., P.L.L. and J.L.F. designed, performed and analysed the NHP study. X.W., J.C.W., B.J.B., M.G., K.M.G., S.M.F. and E.J.R. designed and constructed the ddTMP Mtb kill-switch strain, performed the mutation rate experiment and profiled kill curve in axenic culture. J.C.W. and J.Z. designed and profiled ddTMP degron in mycobacteria. M.Z. and V.D. designed, performed and analysed TMP mouse PK/PD. X.W. and M.G. constructed the TKS strain, performed the mutation rate experiment and profiled kill curve in axenic culture. X.W. performed analysis of mutation rate. Y.J.L. and J.Z. performed analysis on microscopic data. X.W., S.W., J.S., N.C.H., S.E., D.S., S.M.F. and E.J.R. designed, performed and analysed SCID and Rag$^{-/-}$ mouse experiments. X.W., D.M. and E.J.R. wrote the initial manuscript. All authors contributed to the writing and editing of the final manuscript.

## Competing interests

The authors declare no competing interests.

## Additional information

**Extended data** is available for this paper at https://doi.org/10.1038/s41564-024-01913-5.

**Correspondence and requests for materials** should be addressed to Sabine Ehrt, Sarah M. Fortune, Eric J. Rubin or Dirk Schnappinger.

¹Department of Immunology and Infectious Diseases, Harvard T.H. Chan School of Public Health, Boston, MA, USA. ²Department of Microbiology and Immunology, Weill Cornell Medical College, New York, NY, USA. ³Department of Microbiology and Molecular Genetics, University of Pittsburgh School of Medicine, Pittsburgh, PA, USA. ⁴Center for Vaccine Research, University of Pittsburgh, Pittsburgh, PA, USA. ⁵Center for Discovery and Innovation, Hackensack Meridian Health, Nutley, NJ, USA. ⁶Hackensack Meridian School of Medicine, Hackensack Meridian Health, Nutley, NJ, USA. ⁷Department of Pediatrics, Children's Hospital of Pittsburgh of the University of Pittsburgh Medical Center, Pittsburgh, PA, USA. ⁸The Ragon Institute of MGH, MIT and Harvard, Cambridge, MA, USA. ⁹Present address: Center for Veterinary Science, Zhejiang University, Hangzhou, China. ¹⁰These authors contributed equally: Xin Wang, Hongwei Su, Joshua B. Wallach, Jeffrey C. Wagner. ✉e-mail: sae2004@med.cornell.edu; sfortune@hsph.harvard.edu; erubin@hsph.harvard.edu; dis2003@med.cornell.edu

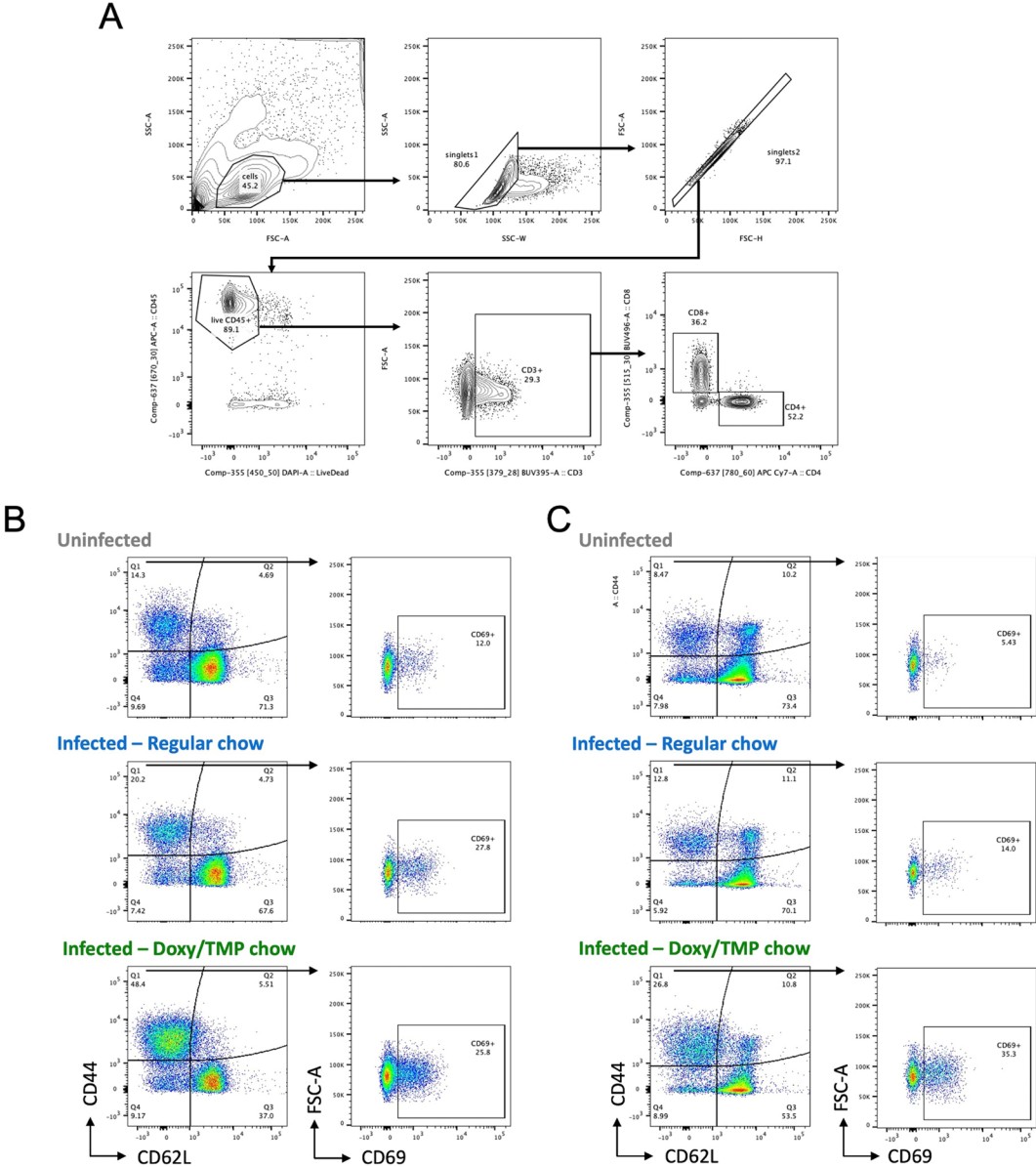

**Extended Data Fig. 1 | Gating strategy for flow cytometry analysis of mouse lung cells. (A)** General gating strategy for CD4 and CD8 T cells. **(B)** Gating strategy for CD4 T cell memory populations. Central memory = CD62L⁺ CD44⁺ (Q2), effector memory = CD62L- CD44⁺ (Q1), resident memory = CD62L⁻ CD44⁺

CD69⁺. **(C)** Gating strategy for CD8 T cell memory populations. Central memory = CD62L⁺ CD44⁺ (Q2), effector memory = CD62L⁻ CD44⁺ (Q1), resident memory = CD62L⁻ CD44⁺ CD69⁺.

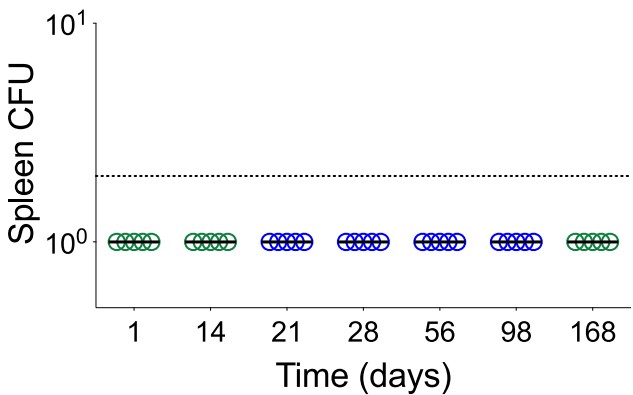

**Extended Data Fig. 2 | Spleen CFU in the SCID experiment.** The SCID mice were fed permissive chow for 14 days, followed by restrictive chow till day 98. After that, the mice received permissive chow till day 168. Spleen CFU were enumerated on 7H11 plates. Each time point include 5 mice. The average and standard deviation are shown on plots. Green indicates mice in permissive condition and blue indicates restrictive condition. The dashed line is the lower limit of detection.

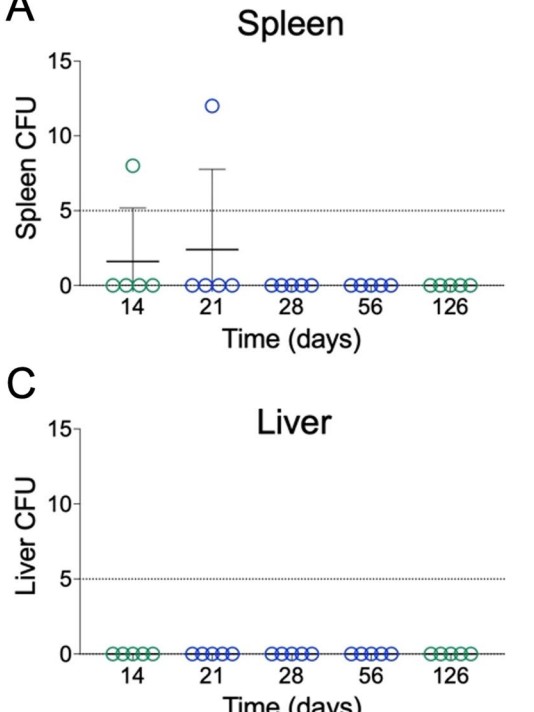

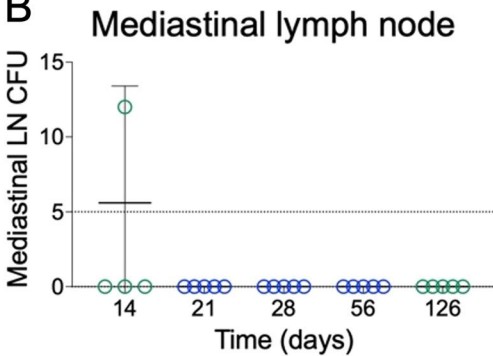

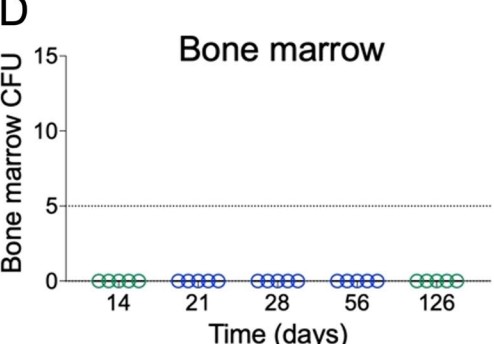

**Extended Data Fig. 3 | TKS strain burden in extrapulmonary tissues in the Rag⁻/⁻ relapse experiment.** The Rag$^{-/-}$ mice were fed permissive chow for 14 days, followed by restrictive chow till day 56. After that, the mice received permissive chow till day 126. The TKS bacteria burden were enumerated in multiple organs, including (**A**) spleen, (**B**) mediastinal lymph node, (**C**) liver and (**D**) bone marrow. Each time point include 5 mice. The average and standard deviation are shown on plots. Green indicates mice in permissive condition and blue indicates restrictive condition. The dashed line is the lower limit of detection.

## Reporting Summary

## Statistics

For all statistical analyses, confirm that the following items are present in the figure legend, table legend, main text, or Methods section.

| n/a | Confirmed | |
|---|---|---|
| ☐ | ☒ | The exact sample size (*n*) for each experimental group/condition, given as a discrete number and unit of measurement |
| ☐ | ☒ | A statement on whether measurements were taken from distinct samples or whether the same sample was measured repeatedly |
| ☐ | ☒ | The statistical test(s) used AND whether they are one- or two-sided *Only common tests should be described solely by name; describe more complex techniques in the Methods section.* |
| ☒ | ☐ | A description of all covariates tested |
| ☐ | ☒ | A description of any assumptions or corrections, such as tests of normality and adjustment for multiple comparisons |
| ☐ | ☒ | A full description of the statistical parameters including central tendency (e.g. means) or other basic estimates (e.g. regression coefficient) AND variation (e.g. standard deviation) or associated estimates of uncertainty (e.g. confidence intervals) |
| ☐ | ☒ | For null hypothesis testing, the test statistic (e.g. *F*, *t*, *r*) with confidence intervals, effect sizes, degrees of freedom and *P* value noted *Give P values as exact values whenever suitable.* |
| ☒ | ☐ | For Bayesian analysis, information on the choice of priors and Markov chain Monte Carlo settings |
| ☒ | ☐ | For hierarchical and complex designs, identification of the appropriate level for tests and full reporting of outcomes |
| ☐ | ☒ | Estimates of effect sizes (e.g. Cohen's *d*, Pearson's *r*), indicating how they were calculated |

*Our web collection on statistics for biologists contains articles on many of the points above.*

## Software and code

Policy information about availability of computer code

| Data collection | No computer code was used in data collection. |
|---|---|
| Data analysis | Data were analyzed and visualized by GraphPad Prism 10. The whole genomic sequencing data was analyzed by Sickle (version 1.33), bwa (version 0.7.17), samtools (version 1.11-9) and VarScan (2.3.9). Flow cytometry was analyzed via FlowJo (version 10.8). Bacteria morphology imaging was analyzed via python package MOMIA (0.0.1). |

For manuscripts utilizing custom algorithms or software that are central to the research but not yet described in published literature, software must be made available to editors and reviewers. We strongly encourage code deposition in a community repository (e.g. GitHub). See the Nature Portfolio guidelines for submitting code & software for further information.

## Data

Policy information about availability of data

All manuscripts must include a data availability statement. This statement should provide the following information, where applicable:
- Accession codes, unique identifiers, or web links for publicly available datasets
- A description of any restrictions on data availability
- For clinical datasets or third party data, please ensure that the statement adheres to our policy

All relevant data generated in this study are present within the manuscript and Supplemental Information. Whole genome sequencing data for escape mutants of dual-lysin strain, ddTMP strain and TKS strain will be available on SRA, and the project and accession numbers will be listed prior to publication.

# Research involving human participants, their data, or biological material

Policy information about studies with human participants or human data. See also policy information about sex, gender (identity/presentation), and sexual orientation and race, ethnicity and racism.

| | |
|---|---|
| Reporting on sex and gender | N/A |
| Reporting on race, ethnicity, or other socially relevant groupings | N/A |
| Population characteristics | N/A |
| Recruitment | N/A |
| Ethics oversight | N/A |

Note that full information on the approval of the study protocol must also be provided in the manuscript.

# Field-specific reporting

Please select the one below that is the best fit for your research. If you are not sure, read the appropriate sections before making your selection.

☒ Life sciences   ☐ Behavioural & social sciences   ☐ Ecological, evolutionary & environmental sciences

For a reference copy of the document with all sections, see nature.com/documents/nr-reporting-summary-flat.pdf

# Life sciences study design

All studies must disclose on these points even when the disclosure is negative.

| | |
|---|---|
| Sample size | No statistical methods were used to pre-determine sample sizes but our sample sizes are similar to those reported in previous publications (PMIDs: 24315099, 34269789). |
| Data exclusions | No data were excluded from analysis. |
| Replication | All experiments contain biological replicates. All growth curve experiments, western blots and flow cytometry studies were repeated >2 times. All replicate experiments reproduced the original results. |
| Randomization | Mice and NHPs used in this study were randomly assigned into groups. |
| Blinding | Microscopy samples were de-identified prior to imaging. Blinding was not applicable due to experiment design, |

# Reporting for specific materials, systems and methods

We require information from authors about some types of materials, experimental systems and methods used in many studies. Here, indicate whether each material, system or method listed is relevant to your study. If you are not sure if a list item applies to your research, read the appropriate section before selecting a response.

## Materials & experimental systems

| n/a | Involved in the study |
|---|---|
| ☐ | ☒ Antibodies |
| ☒ | ☐ Eukaryotic cell lines |
| ☒ | ☐ Palaeontology and archaeology |
| ☐ | ☒ Animals and other organisms |
| ☒ | ☐ Clinical data |
| ☒ | ☐ Dual use research of concern |
| ☒ | ☐ Plants |

## Methods

| n/a | Involved in the study |
|---|---|
| ☒ | ☐ ChIP-seq |
| ☐ | ☒ Flow cytometry |
| ☒ | ☐ MRI-based neuroimaging |

# Antibodies

| | |
|---|---|
| Antibodies used | anti-myc (71D40, cell signaling); anit-RpoB (8RB13, cell signaling); anti-CD16/32 (101302, Biolegend), anti-CD69-BB700 (HI.2F3, BD Biosciences); anti-CD44-FITC (IM7, Biolegend), anti-CD62L-BV650 (MEL14, BD Biosciences), anti-CD11a-BV605 (2D7, 424 BD Biosciences), anti-KLRG1-PE-Cy7 (2FI, Biolegend), anti-PD-1-BV421 (29F.1AI2, Biolegend), anti-CD8-425-BUV496 (53-6.7, BD |

Biosciences), **anti-CD3-BUV395** (17A2, BD Biosciences), **anti-CD4-APC-H7 426** (GKI.5, BD Biosciences), **anti-CD45-APC** (30-Fll, Biolegend), **IRDye® 680RD goat IgG (H + L) anti-mouse** (926-68070, LICORbio), **IRDye® 800CW goat IgG (H + L) anti-rabbit** (926-68071, LICORbio).

| Validation | Antibodies were validated by the manufacturer. We also validated antibody via protein size for western blot and positive controls in flow cytometry. |

# Animals and other research organisms

Policy information about studies involving animals; ARRIVE guidelines recommended for reporting animal research, and Sex and Gender in Research

| Laboratory animals | Mouse: C57BL/6J mice (The Jackson Laboratory, 000664); B6.Cg-Prkdcscid/SzJ mice (The Jackson Laboratory, 001913); B6.129S7-Ragltm1Mom/J mice (The Jackson Laboratory, 002216). All mice are 6- to 8-week old.<br>Nonhuman primates: 6-9 years old Mauritian cynomolgus macaques (Maccaca Fascicularis), obtained from Bioculture Mauritius. |

| Wild animals | No wild animals were used in this study. |

| Reporting on sex | All mice were female. All NHP were male. |

| Field-collected samples | No field collected samples were used in this study. |

| Ethics oversight | For mouse experiments, animal care and experimental procedures were conducted with the approval of the Institutional Animal Care and Use Committees (IACUC) of Harvard Medical School, Harvard T.H. Chan School of Public Health and Weill Cornell Medicine.<br><br>For NHP studies, all experimental manipulations, procedures, protocols, and care of the animals were approved by the University of Pittsburgh School of Medicine Institutional Animal Care and Use Committee (IACUC). The protocol assurance number for our IACUC is A3187-01. The specific protocol approval number is 18124087. The University of Pittsburgh's IACUC adheres to national guidelines established in the Animal Welfare Act (7 U.S.C. Sections 2131–2159) and the Guide for the Care and Use of Laboratory Animals (eighth edition), as mandated by the U.S. Public Health Service Policy. |

Note that full information on the approval of the study protocol must also be provided in the manuscript.

# Plants

| Seed stocks | *Report on the source of all seed stocks or other plant material used. If applicable, state the seed stock centre and catalogue number. If plant specimens were collected from the field, describe the collection location, date and sampling procedures.* |

| Novel plant genotypes | *Describe the methods by which all novel plant genotypes were produced. This includes those generated by transgenic approaches, gene editing, chemical/radiation-based mutagenesis and hybridization. For transgenic lines, describe the transformation method, the number of independent lines analyzed and the generation upon which experiments were performed. For gene-edited lines, describe the editor used, the endogenous sequence targeted for editing, the targeting guide RNA sequence (if applicable) and how the editor was applied.* |

| Authentication | *Describe any authentication procedures for each seed stock used or novel genotype generated. Describe any experiments used to assess the effect of a mutation and, where applicable, how potential secondary effects (e.g. second site T-DNA insertions, mosiacism, off-target gene editing) were examined.* |

# Flow Cytometry

## Plots

Confirm that:

☒ The axis labels state the marker and fluorochrome used (e.g. CD4-FITC).

☒ The axis scales are clearly visible. Include numbers along axes only for bottom left plot of group (a 'group' is an analysis of identical markers).

☒ All plots are contour plots with outliers or pseudocolor plots.

☒ A numerical value for number of cells or percentage (with statistics) is provided.

## Methodology

| Sample preparation | Mtb flow cytometry: Mtb cells were fixed in 2% paraformaldehyde overnight and removed from the BSL-3 facility. 331 Fixed bacilli were quenched with 200 mM Tris-HCl (pH 7.5) for 5 minutes at room temperature 332 and resuspended in PBST buffer (1x PBS with 0.1% Triton X-100). To suppress signals from 333 noise or cell debris, two event triggers (thresholds) on forward scatter peak height (FSC-H >1.5) 334 and side scatter area (SSC-A > 1.0) were used upon recording. |

For lung cell flow cytometry: Lung cell suspensions were washed with PBS twice, stained for viability with the live/dead dye Zombie UV (Biolegend, 423107) for 10 min at 4°C, washed with Cell Staining Buffer (BioLegend 420201) and incubated with Fc block (purified anti-mouse CD16/CD32 antibody, Biolegend 101302) at 1:200 in Cell Staining Buffer for 10 min at 4°C. After one wash in Cell Staining Buffer, cells were incubated with fluorochrome-conjugated monoclonal antibodies (mAbs) diluted at 1:200 into a 1:3 solution of Brilliant Staining Buffer (BD 563794): Cell Staining Buffer, for 45 min at 4°C. Cells were then washed twice in Cell Staining Buffer and fixed with Fixation Buffer (Biolegend 420801) for 30 min at 4°C. In all incubation steps cells were protected from light. Fluorescence minus one (FMO) controls were stained alongside samples.

| Instrument | Mtb flow cytometry: MACSQuant Analyzer 10 flow cytometer (Miltenyi Biotec)<br>Lung flow cytometry: FACSymphony A5 Cell Analyzer (BD Biossciences) |
|---|---|
| Software | FlowJo v10.8 (BD Life Sciences) was used for flow cytometry analysis. |
| Cell population abundance | The final cell/bacteria population was 0.5 to 20% of total events. |
| Gating strategy | Mtb flow cytometry: To suppress signals from noise or cell debris, two event triggers (thresholds) on forward scatter peak height (FSC-H >1.5) and side scatter area (SSC-A > 1.0) were used upon recording. To remove cellular aggregation, stringent gate settings were manually defined via FlowJo v. 10.8 to exclude events with strongly correlated forward scatter area (FSC-A) and SSC-A measures (large and compact particles), as well as events with disproportional FSC-A and FSC-H measures (morphological outliers). After event filtration, the log10-transformed red fluorescence intensity peak height (denoted TdTomato-A) was used to represent the abundance of intracellular red fluorescence protein.<br><br>Lung cytometry: The gating strategy is shown in supplement figure 1. |

☒ Tick this box to confirm that a figure exemplifying the gating strategy is provided in the Supplementary Information.

