## [Peer Review File · Nature Microbiology]

Engineered *Mycobacterium tuberculosis* triple kill switch strain provides controlled tuberculosis infection in animal models

Corresponding Author: Dr Eric Rubin

Version 0:

Decision Letter:

10th August 2023

Dear Dr Ruben,

thank you very much for your enquiry about submitting a manuscript to Nature Microbiology and my apologies for the time taken to get back to you. It is a busy time at the journal at the moment and a number of the team have been busy with travel for various reasons meaning our response times are a longer than usual.

I've now had a chance to discuss your work with my colleagues, and we think that it sounds very interesting as there is obviously huge translational potential for such a tool and hopefully it could also yield major biological insight into disease progression and *M. tuberculosis* in vivo biology. However, we would like to see the data prior to making a decision on whether we would send the paper out to review. With this in mind, I would like to invite you to submit the full manuscript as soon as it is ready, so that myself and the rest of the editorial team can take a look at the data and get back to you with a decision.

If this is acceptable to you, you can submit the complete manuscript using the link below:

Link Redacted

If you have any questions, please feel free to contact me.

Yours sincerely

Version 1:

Reviewer comments:

Reviewer #1

(Remarks to the Author)

Here Wang et al describe the generation of a strain of *M. tuberculosis* for use in human challenge studies. Initially, the authors produce a strain that is induced to express two different phage lysins upon withdrawal of tetracyclines. However, the escape frequency of this strain failed to meet the authors strict criteria and in vivo experiments in mice and macaques showed that or rate of decline after withdrawal of tetracyclines was slower than hoped. Therefore, the authors added a third kill switch. This new strain was additional induced to degrade the essential bacterial protein NadE upon removal of trimethoprim. This new triple kill strain has a strikingly low escape rate and dies much faster in vitro and in mice when deprived of the antibiotics. Moreover, the authors were unable to reactivate any bacteria in SCID mice. The authors are to be congratulated for the numerous thoughtful technical approaches employed to create a strain of *Mtb* seemingly safe to be administered to humans in controlled challenge studies. This is an important advance that is widely anticipated by the field. I only have a few minor comments.

1. How many times were the mouse model experiments repeated? While NHP experiments are not typically repeated for practical, cost and ethical reasons, experiments done in mouse models must be conducted at least twice. This is particularly important given the current focus on reproducibility in science and the intent of this paper to report a product intended for human use. If these experiments were only done once, the authors should point this out as a limitation and justify why a repeat experiment was not necessary. For example, it may be considered acceptable as there are multiple measurements in a kinetics analysis, although some investigators may consider even this not sufficient reason for the lack of a second experiment to demonstrate reproducibility. I have made the same comment in my Review of the accompanying paper by Smith et al.

2. Is it possible that individuals challenged with the triple kill strain might be immunized against the phage lysins? Perhaps its unlikely, but I wonder if antibodies against phage lysins could impair the lysis of extracellular bacteria. Although not required to support the authors conclusions about safety, it would be good to know if antibody responses against the lysins were detected in the macaques.

Reviewer #2

(Remarks to the Author)

The study explores the development of an *M. tuberculosis* strain for potential use in human challenge experiments. The authors investigate multiple kill switches regulated by exogenous compounds tetracycline and trimethoprim individually and in combination. The strains demonstrate immunogenicity and antibiotic susceptibility similar to wild-type *Mtb* under permissive conditions. They exhibit rapid elimination in the absence of external compounds, suggesting potential safety and effectiveness for a human challenge model. Notably, the three-kill-switch strain shows promising characteristics, including a minimal escape rate and no relapse in a mouse model. The authors display meticulous attention to detail and creativity in constructing kill switches, underscoring their commitment to precision in experimental design.

Comments:

Line 248: The others set their goal for an escape rate of $\leq 10-12$. How did the authors arrive at this rate? Providing context for this goal within this magnitude will help the reader appreciate the achieved calculated mutation rate of $\leq 10-15$.

The mutations of the dual lysin mutants should also be visually represented similarly to the mutants of the TMP kill switch in Figure 3F.

Line 127: The reference used (reference 23) seems unsuitable for this statement, as the study by Forgacs et al tested clinical *Mtb* isolates against TMP in combination with sulfamethoxazole and detected susceptibility.

Lines 103-114: It should be stated in the text (and Fig. 1) how much time passed between the last doxy supplementation and the necropsies of the animals to allow the reader to put those experiments into perspective. Only the method section suggests it might have been 6 weeks after the 2 weeks and 6 weeks challenge.

Line 452: The method section of the NHP infection study is missing the quantity of administered Doxycycline. The outstandingly low escape rate of 10-12 of the TKS strain is achieved by the combination of Doxy and TMP kill switches. Each kill switch by itself does not fulfill the strict safety guidelines. The possibility of unevenly distributed inducers in the host and complex environments of *Mtb* infection sites could be discussed. Exposure of the TKS strain to only one inducer could allow the strain to escape stepwise, or this might not be possible because of sufficiently high concentrations of Doxy and TMP.

For consistency, please identify the dashed line in Fig. 1B/E, Fig. 5B/C/I.

Indicate the organism in Fig. 5E similarly to panels A-D.

Fig. 5 text: Please consistently name the strains. The "combo kill switch strain" should be the TKS strain.

Reviewer #3

(Remarks to the Author)

This manuscript by Wang et al describes generation and characterization of a strain of *Mtb* containing three kill switches. The investigators first generated a tetracycline-dependent dual lysin *Mtb* strain and then engineered a strain to combine two lysin switches with NadE-ddTMP to yield a triple-kill switch (TKS) *Mtb* H37Rv strain. The investigators characterize growth kinetics, antibiotic susceptibility, and immune profiling of the TKS strain in mice. The experiments described are well controlled and of high rigor and create a strong foundation for future studies to continue characterization of the TKS strain in evaluating its potential utility for development and application in human infection model (CHIM) studies.

The studies are of high importance given the current lack of an effective vaccine against pulmonary TB in adolescents and adults and lack of well-defined immune correlates of protection against TB. Development of a safe controlled human infection model (CHIM) for *Mtb* would be advantageous to the field of TB vaccinology and treatment. The investigators use SCID mice to demonstrate lack of growth of the TKS strain in lung and spleen after return to doxy and TMP supplementation. Rigorous safety studies will need to be conducted prior to use of engineered *Mtb* strains for use in CHIM studies, and the SCID experiments presented in this manuscript provide important preliminary data towards establishing a safety profile of this strain for potential use in humans. Clarification on the comments below would strengthen the conclusions that can be drawn from the manuscript.

Comments:

1. The Abstract (lines 31-32) and conclusion of the Discussion section (lines 281-282) stating that the immunogenicity of TKS strain is comparable to the wild-type strain is not strongly supported by the data presented in this manuscript. Minimal immune profiling data are shown (only one time point was evaluated, with no evaluation of either innate immune responses or *Mtb*-specific T cell responses). Immune profiling data of TKS-infected mice are presented along with uninfected mice, although there are no immune profiling data comparing TKS with wt *Mtb*. Either the Abstract/Discussion summary and conclusions should be modified to more accurately reflect the data presented in the manuscript, or additional experiments provided to demonstrate the direct comparison of immunogenicity of TKS with wt *Mtb*.

2. Figure 5, panels D-G: it is not clear what condition is represented by the grey bars. The implication from the Results section (line 217) is that the grey bars are uninfected mice, although this should be clarified in Figure 5 and in the figure legend.
3. Lung CD4 and CD8 T cells were evaluated at day 84 post-infection with the TKS strain. T cells were evaluated for activation and effector T cell populations, although Mtb-specific immunogenicity was not evaluated. Only one time point was evaluated for immunogenicity (d84), although it would be useful to evaluate immunogenicity at earlier time points, as well as comparison with mice infected with wt H37Rv.
4. Line 217 indicates the T cell response at d84 in mice under restrictive conditions were distinguishable from uninfected mice, although this conclusion is not strongly supported by the data in Figure 5D-E. There are no differences in CD4 T cell responses between uninfected and mice under restrictive conditions (Figure 5D, E) and only modestly higher frequencies of effector CD8 T cell responses under restrictive conditions, compared with uninfected mice (Figure 5G). This conclusion should be modified to reflect the data in Figure 5 more accurately.
5. In experiments of TKS infection of SCID mice, lung and spleen were cultured for CFU, but no other tissues were cultured. Viable bacteria may be present in either LN, BM, or other tissues. The number of SCID mice evaluated was small (n=5 for each time point), and although the experiments performed provide promising data, it will be important to demonstrate eradication of viable bacteria in a more comprehensive manner and in a larger number of animals in future studies characterizing the TKS strain prior to use in humans.
6. Infection of SCID mice is important to demonstrate lack of reemergence of bacterial growth, although the number of SCID mice per time point is small (n=5) and warrants repeating with larger numbers in future studies (not necessarily in this manuscript) to more robustly characterize the TKS strain prior to use in CHIM studies. The authors could consider expanding on this point in the Discussion section.
7. The TKS strain requires presence of both aTc and TMP, provided to mice in the form of supplemented chow. The authors suggest that the TKS strain could be used in human challenge studies, although the feasibility and practicality of conducting PK/PD studies to establish necessary levels of aTc and TMP in humans to ensure safety of the TKS strain in human challenge studies is not clear and could be further acknowledged as a limitation in the Discussion section.

Decision Letter:

29th January 2024

Dear Dr Rubin,

Thank you for your patience while your manuscript "Development of an Engineered Mycobacterium tuberculosis Strain for a Safe and Effective Tuberculosis Human Challenge Model" was under peer-review at Nature Microbiology. It has now been seen by 3 referees, whose expertise and comments you will find at the end of this email. Although they find your work of some potential interest, they have raised a number of concerns that will need to be addressed before we can consider publication of the work in Nature Microbiology.

In particular, you will see that several referees raise questions over the number of independent biological replicates carried out for the mouse experiments, as well as asking that immunogenicity be compared between infections with the WT M. tuberculosis strain and the mutant strain, to support statements that similar responses are elicited. Referee #3 also raised concerns that more data were needed to support the claims of bacterial clearance in SCID mice, including the analysis of additional tissues. Although we recognise that these are lengthy experiments, given the potential importance of this study we feel that it is critical that a revised study would satisfy these points, including additional in vivo analyses as requested in mice for the key experiments detailed by the referees. The rest of the referees' reports are clear and the remaining issues should be straightforward to address.

Should further experimental data allow you to address these criticisms, we would be happy to look at a revised manuscript.

Please include a data availability statement as a separate section after Methods but before references, under the heading "Data Availability". This section should inform readers about the availability of the data used to support the conclusions of your study. This information includes accession codes to public repositories (data banks for protein, DNA or RNA sequences, microarray, proteomics data etc...), references to source data published alongside the paper, unique identifiers such as URLs to data

repository entries, or data set DOIs, and any other statement about data availability. At a minimum, you should include the following statement: "The data that support the findings of this study are available from the corresponding author upon request", mentioning any restrictions on availability. If DOIs are provided, we also strongly encourage including these in the Reference list (authors, title, publisher (repository name), identifier, year). For more guidance on how to write this section please see: <http://www.nature.com/authors/policies/data/data-availability-statements-data-citations.pdf>

* If you have not done so already we suggest that you begin to revise your manuscript so that it conforms to our Article format instructions at <http://www.nature.com/nmicrobiol/info/final-submission>. Refer also to any guidelines provided in this letter.

When submitting the revised version of your manuscript, please pay close attention to our [href="https://www.nature.com/nature-portfolio/editorial-policies/image-integrity">Digital Image Integrity Guidelines. and to the following points below:](https://www.nature.com/nature-portfolio/editorial-policies/image-integrity)

Link Redacted

Note: This url links to your confidential homepage and associated information about manuscripts you may have submitted or be reviewing for us. If you wish to forward this e-mail to co-authors, please delete this link to your homepage first.

Nature Microbiology is committed to improving transparency in authorship. As part of our efforts in this direction, we are now requesting that all authors identified as 'corresponding author' on published papers create and link their Open Researcher and Contributor Identifier (ORCID) with their account on the Manuscript Tracking System (MTS), prior to acceptance. This applies to primary research papers only. ORCID helps the scientific community achieve unambiguous attribution of all scholarly contributions. You can create and link your ORCID from the home page of the MTS by clicking on 'Modify my Springer Nature account'. For more information please visit [please visit www.springernature.com/orcid](http://www.springernature.com/orcid).

If you wish to submit a suitably revised manuscript we would hope to receive it within 6 months. If you cannot send it within this time, please let us know. We will be happy to consider your revision, even if a similar study has been accepted for publication at Nature Microbiology or published elsewhere (up to a maximum of 6 months).

Yours sincerely,

Reviewer Expertise:

Referee #1: Mtb, immunology/T cells, vaccine responses

Referee #2: Mtb genetics, drug/therapeutic development

Referee #3: TB/HIV immunology, vaccine development

Reviewer Comments:

Reviewer #1 (Remarks to the Author):

Here Wang et al describe the generation of a strain of M. tuberculosis for use in human challenge studies. Initially, the authors produce a strain that is induced to express two different phage lysins upon withdrawal of tetracyclines. However, the escape frequency of this strain failed to meet the authors strict criteria and in vivo experiments in mice and macaques showed that or rate of decline after withdrawal of tetracyclines was slower than hoped. Therefore, the authors added a third kill switch. This new

strain was additionally induced to degrade the essential bacterial protein NadE upon removal of trimethoprim. This new triple kill strain has a strikingly low escape rate and dies much faster *in vitro* and in mice when deprived of the antibiotics. Moreover, the authors were unable to reactivate any bacteria in SCID mice. The authors are to be congratulated for the numerous thoughtful technical approaches employed to create a strain of Mtb seemingly safe to be administered to humans in controlled challenge studies. This is an important advance that is widely anticipated by the field. I only have a few minor comments.

1. How many times were the mouse model experiments repeated? While NHP experiments are not typically repeated for practical, cost and ethical reasons, experiments done in mouse models must be conducted at least twice. This is particularly important given the current focus on reproducibility in science and the intent of this paper to report a product intended for human use. If these experiments were only done once, the authors should point this out as a limitation and justify why a repeat experiment was not necessary. For example, it may be considered acceptable as there are multiple measurements in a kinetics analysis, although some investigators may consider even this not sufficient reason for the lack of a second experiment to demonstrate reproducibility. I have made the same comment in my Review of the accompanying paper by Smith et al.

2. Is it possible that individuals challenged with the triple kill strain might be immunized against the phage lysins? Perhaps it's unlikely, but I wonder if antibodies against phage lysins could impair the lysis of extracellular bacteria. Although not required to support the authors' conclusions about safety, it would be good to know if antibody responses against the lysins were detected in the macaques.

Reviewer #2 (Remarks to the Author):

The study explores the development of an *M. tuberculosis* strain for potential use in human challenge experiments. The authors investigate multiple kill switches regulated by exogenous compounds tetracycline and trimethoprim individually and in combination. The strains demonstrate immunogenicity and antibiotic susceptibility similar to wild-type Mtb under permissive conditions. They exhibit rapid elimination in the absence of external compounds, suggesting potential safety and effectiveness for a human challenge model. Notably, the three-kill-switch strain shows promising characteristics, including a minimal escape rate and no relapse in a mouse model. The authors display meticulous attention to detail and creativity in constructing kill switches, underscoring their commitment to precision in experimental design.

Comments:

Line 248: The authors set their goal for an escape rate of $\leq 10^{-12}$. How did the authors arrive at this rate? Providing context for this goal within this magnitude will help the reader appreciate the achieved calculated mutation rate of $\leq 10^{-15}$.

The mutations of the dual lysin mutants should also be visually represented similarly to the mutants of the TMP kill switch in Figure 3F.

Line 127: The reference used (reference 23) seems unsuitable for this statement, as the study by Forgacs et al tested clinical Mtb isolates against TMP in combination with sulfamethoxazole and detected susceptibility.

Lines 103-114: It should be stated in the text (and Fig. 1) how much time passed between the last doxy supplementation and the necropsies of the animals to allow the reader to put those experiments into perspective. Only the method section suggests it might have been 6 weeks after the 2 weeks and 6 weeks challenge.

Line 452: The method section of the NHP infection study is missing the quantity of administered Doxycycline. The outstandingly low escape rate of 10^{-12} of the TKS strain is achieved by the combination of Doxy and TMP kill switches. Each kill switch by itself does not fulfill the strict safety guidelines. The possibility of unevenly distributed inducers in the host and complex environments of Mtb infection sites could be discussed. Exposure of the TKS strain to only one inducer could allow the strain to escape stepwise, or this might not be possible because of sufficiently high concentrations of Doxy and TMP.

For consistency, please identify the dashed line in Fig. 1B/E, Fig. 5B/C/I.

Indicate the organism in Fig. 5E similarly to panels A-D.

Fig. 5 text: Please consistently name the strains. The "combo kill switch strain" should be the TKS strain.

Reviewer #3 (Remarks to the Author):

This manuscript by Wang et al describes generation and characterization of a strain of Mtb containing three kill switches. The investigators first generated a tetracycline-dependent dual lysin Mtb strain and then engineered a strain to combine two lysin switches with NadE-ddTMP to yield a triple-kill switch (TKS) Mtb H37Rv strain. The investigators characterize growth kinetics, antibiotic susceptibility, and immune profiling of the TKS strain in mice. The experiments described are well controlled and of high rigor and create a strong foundation for future studies to continue characterization of the TKS strain in evaluating its potential utility for development and application in human infection model (CHIM) studies.

The studies are of high importance given the current lack of an effective vaccine against pulmonary TB in adolescents and adults and lack of well-defined immune correlates of protection against TB. Development of a safe controlled human infection model (CHIM) for Mtb would be advantageous to the field of TB vaccinology and treatment. The investigators use SCID mice to demonstrate lack of growth of the TKS strain in lung and spleen after return to doxy and TMP supplementation. Rigorous safety

studies will need to be conducted prior to use of engineered Mtb strains for use in CHIM studies, and the SCID experiments presented in this manuscript provide important preliminary data towards establishing a safety profile of this strain for potential use in humans. Clarification on the comments below would strengthen the conclusions that can be drawn from the manuscript.

Comments:

1. The Abstract (lines 31-32) and conclusion of the Discussion section (lines 281-282) stating that the immunogenicity of TKS strain is comparable to the wild-type strain is not strongly supported by the data presented in this manuscript. Minimal immune profiling data are shown (only one time point was evaluated, with no evaluation of either innate immune responses or Mtb-specific T cell responses). Immune profiling data of TKS-infected mice are presented along with uninfected mice, although there are no immune profiling data comparing TKS with wt Mtb. Either the Abstract/Discussion summary and conclusions should be modified to more accurately reflect the data presented in the manuscript, or additional experiments provided to demonstrate the direct comparison of immunogenicity of TKS with wt Mtb.
2. Figure 5, panels D-G: it is not clear what condition is represented by the grey bars. The implication from the Results section (line 217) is that the grey bars are uninfected mice, although this should be clarified in Figure 5 and in the figure legend.
3. Lung CD4 and CD8 T cells were evaluated at day 84 post-infection with the TKS strain. T cells were evaluated for activation and effector T cell populations, although Mtb-specific immunogenicity was not evaluated. Only one time point was evaluated for immunogenicity (d84), although it would be useful to evaluate immunogenicity at earlier time points, as well as comparison with mice infected with wt H37Rv.
4. Line 217 indicates the T cell response at d84 in mice under restrictive conditions were distinguishable from uninfected mice, although this conclusion is not strongly supported by the data in Figure 5D-E. There are no differences in CD4 T cell responses between uninfected and mice under restrictive conditions (Figure 5D, E) and only modestly higher frequencies of effector CD8 T cell responses under restrictive conditions, compared with uninfected mice (Figure 5G). This conclusion should be modified to reflect the data in Figure 5 more accurately.
5. In experiments of TKS infection of SCID mice, lung and spleen were cultured for CFU, but no other tissues were cultured. Viable bacteria may be present in either LN, BM, or other tissues. The number of SCID mice evaluated was small (n=5 for each time point), and although the experiments performed provide promising data, it will be important to demonstrate eradication of viable bacteria in a more comprehensive manner and in a larger number of animals in future studies characterizing the TKS strain prior to use in humans.
6. Infection of SCID mice is important to demonstrate lack of reemergence of bacterial growth, although the number of SCID mice per time point is small (n=5) and warrants repeating with larger numbers in future studies (not necessarily in this manuscript) to more robustly characterize the TKS strain prior to use in CHIM studies. The authors could consider expanding on this point in the Discussion section.
7. The TKS strain requires presence of both aTc and TMP, provided to mice in the form of supplemented chow. The authors suggest that the TKS strain could be used in human challenge studies, although the feasibility and practicality of conducting PK/PD studies to establish necessary levels of aTc and TMP in humans to ensure safety of the TKS strain in human challenge studies is not clear and could be further acknowledged as a limitation in the Discussion section.

Version 2:

Reviewer comments:

Reviewer #1

(Remarks to the Author)

The new data in the Rag-/- mice has nicely confirmed the previous results, and all of my concerns have been addressed.

Reviewer #2

(Remarks to the Author)

Thank you for the detailed response to the review comments. I appreciate the effort taken to address all the raised questions. The rebuttal document has sufficiently clarified the issues, and I am satisfied with the answers provided.

I acknowledge that the authors' internal goal of achieving a mutation rate with an escape rate of $\leq 10^{-12}$ was influenced by technical limitations. Although this was not fully clarified in the manuscript, the published rebuttal document will provide an adequate explanation.

I noticed a slight inconsistency in how data points below the limit of detection are represented. In Figure 1, the data points are

placed directly on the "limit of detection" line, while in Figures 4 and 5, some points fall below it. If both cases are intended to convey that no bacteria were detectable, I would suggest opting for a consistent representation.

Reviewer #3

(Remarks to the Author)

The authors have carefully and thoughtfully addressed each of the comments raised in the initial review of this manuscript. The authors have repeated the experiments with Rag-/- mice, which is a further strength of the revised manuscript. Well done to the authors on an excellent study. I have no further comments or concerns to be addressed.

Decision Letter:

Our ref: NMICROBIOL-23081984B

31st October 2024

Dear Eric,

Thank you for submitting your revised manuscript "Development of an Engineered Mycobacterium tuberculosis Strain for a Safe and Effective Tuberculosis Human Challenge Model" (NMICROBIOL-23081984B). It has now been seen by the original referees and their comments are below. The reviewers find that the paper has improved in revision, and therefore we'll be happy in principle to publish it in Nature Microbiology, pending minor revisions to satisfy the referees' final requests and to comply with our editorial and formatting guidelines.

We are now performing detailed checks on your paper and will send you a checklist detailing our editorial and formatting requirements in about two weeks. Please do not upload the final materials and make any revisions until you receive this additional information from us.

Thank you again for your interest in Nature Microbiology Please do not hesitate to contact me if you have any questions.

Best wishes,

Reviewer #1 (Remarks to the Author):

The new data in the Rag-/- mice has nicely confirmed the previous results, and all of my concerns have been addressed.

Reviewer #2 (Remarks to the Author):

Thank you for the detailed response to the review comments. I appreciate the effort taken to address all the raised questions. The rebuttal document has sufficiently clarified the issues, and I am satisfied with the answers provided.

I acknowledge that the authors' internal goal of achieving a mutation rate with an escape rate of $\leq 10^{-12}$ was influenced by technical limitations. Although this was not fully clarified in the manuscript, the published rebuttal document will provide an adequate explanation.

I noticed a slight inconsistency in how data points below the limit of detection are represented. In Figure 1, the data points are placed directly on the "limit of detection" line, while in Figures 4 and 5, some points fall below it. If both cases are intended to convey that no bacteria were detectable, I would suggest opting for a consistent representation.

Reviewer #3 (Remarks to the Author):

The authors have carefully and thoughtfully addressed each of the comments raised in the initial review of this manuscript. The authors have repeated the experiments with Rag-/- mice, which is a further strength of the revised manuscript. Well done to the authors on an excellent study. I have no further comments or concerns to be addressed.

Version 3:

Decision Letter:

12th December 2024

Dear Dr Rubin,

I am pleased to accept your Article "Engineered Mycobacterium tuberculosis triple kill switch strain provides controlled tuberculosis infection in animal models" for publication in Nature Microbiology. Thank you for having chosen to submit your work to us and many congratulations.

Please note that *Nature Microbiology* is a Transformative Journal (TJ). Authors may publish their research with us through the traditional subscription access route or make their paper immediately open access through payment of an article-processing charge (APC). Authors will not be required to make a final decision about access to their article until it has been accepted. [Find out more about Transformative Journals](https://www.springernature.com/gp/open-research/transformative-journals)

Authors may need to take specific actions to achieve [compliance](https://www.springernature.com/gp/open-research/funding/policy-compliance-faqs) with funder and institutional open access mandates. If your research is supported by a funder that requires immediate open access (e.g. according to [Plan S principles](https://www.springernature.com/gp/open-research/plan-s-compliance)) then you should select the gold OA route, and we will direct you to the compliant route where possible. For authors selecting the subscription publication route, the journal's standard licensing terms will need to be accepted, including [self-archiving policies](https://www.nature.com/nature-portfolio/editorial-policies/self-archiving-and-license-to-publish). Those licensing terms will supersede any other terms that the author or any third party may assert apply to any version of the manuscript.

With kind regards,

P.S. Click on the following link if you would like to recommend Nature Microbiology to your librarian
<http://www.nature.com/subscriptions/recommend.html#forms>

** Visit the Springer Nature Editorial and Publishing website at http://editorial-jobs.springernature.com?utm_source=ejP_NMicro_email&utm_medium=ejP_NMicro_email&utm_campaign=ejp_NMicro for more information about our career opportunities. If you have any questions please click [here](mailto:editorial.publishing.jobs@springernature.com).

Response to Reviewers

We thank the reviewers for the thorough assessment of our manuscript and the insightful feedback. We have performed additional experiments and analyses in order to address these comments. Below we included responses to each point by the reviewers in blue.

Reviewer 1:

Here Wang et al describe the generation of a strain of *M. tuberculosis* for use in human challenge studies. Initially, the authors produce a strain that is induced to express two different phage lysins upon withdrawal of tetracyclines. However, the escape frequency of this strain failed to meet the authors strict criteria and in vivo experiments in mice and macaques showed that or rate of decline after withdrawal of tetracyclines was slower than hoped. Therefore, the authors added a third kill switch. This new strain was additional induced to degrade the essential bacterial protein NadE upon removal of trimethoprim. This new triple kill strain has a strikingly low escape rate and dies much faster in vitro and in mice when deprived of the antibiotics. Moreover, the authors were unable to reactivate any bacteria in SCID mice. The authors are to be congratulated for the numerous thoughtful technical approaches employed to create a strain of *Mtb* seemingly safe to be administered to humans in controlled challenge studies. This is an important advance that is widely anticipated by the field. I only have a few minor comments.

1. How many times were the mouse model experiments repeated? While NHP experiments are not typically repeated for practical, cost and ethical reasons, experiments done in mouse models must be conducted at least twice. This is particularly important given the current focus on reproducibility in science and the intent of this paper to report a product intended for human use. If these experiments were only done once, the authors should point this out as a limitation and justify why a repeat experiment was not necessary. For example, it may be considered acceptable as there are multiple measurements in a kinetics analysis, although some investigators may consider even this not sufficient reason for the lack of a second experiment to demonstrate reproducibility. I have made the same comment in my Review of the accompanying paper by Smith et al.

Reply: We appreciate the reviewer's comments on mouse data reproducibility. The mouse experiments in the manuscript (original Fig. 1D and Fig. 5) were only performed once, however the bacterial burden was sampled at multiple time points for the kinetics study. To address the reviewer's concern, we repeated the relapse study in *Rag*^{-/-} mouse (B6.129S7-*Rag1*^{tm1Mom}/J, JAX002216). Although the experiment was not repeated with the SCID mouse as in the original manuscript (original Figure 5H and 5I), we believe that using another immunodeficient mouse enhances the robustness of our findings.

In the *Rag*^{-/-} mouse relapse study (**ReFig. 1A**), we infected mice via aerosol with 50-100 D1 CFU with the TKS strain. The infection was allowed to establish for 2 weeks with doxy/TMP chow, and we switched to regular chow since D14 for clearance. Instead of the longer time in the original manuscript, (10 weeks), we used a more rigorous 6 weeks of feeding supplements. We switched to permissive chow at D56 for 10 weeks until harvest at D126.

In the new Rag^{-/-} relapse study, pulmonary TKS burden reached 2500 CFU at D14, which is at a similar level as the previous SCID mouse study (5000 CFU) (**ReFig. 1B**). The average pulmonary clearance rate is 0.89 log CFU per week, which is slightly slower than the clearance rate in the SCID study. At day 28, the lung from one of five mice was sterile and all mice sterilized by day 56 (**ReFig. 1B**). In addition to lung, we also examined the disseminated bacterial burden in other organs, including mediastinal lymph node, spleen, liver and bone marrow (**ReFig. 1C to 1F**). At the end of the establishment phase (D14), most mice were free from dissemination except for 2 mice. One mouse had 12 CFU in the mediastinal lymph node and 8 CFU in the spleen, and the other had 16 CFU in the mediastinal lymph node only. During the clearance phase (D14 to D56), all organs were sterile except in one mouse at D21 with 12 CFU in the spleen. After re-feeding with doxy/TMP chow for 10 weeks, all five mice at day 126 were completely sterile in all examined organs including lung, spleen, liver, mediastinal lymph node and bone marrow (**ReFig. 1B to 1F**). This result confirms that we do not observe relapse in immunodeficient mouse models even after relatively short clearance times. These data are included in the manuscript as Figures 5J and 5K, and Supplemental Figure 3. Accordingly, we added the text below in the lines 233 to 247:

To assess the potential for relapse under stringent conditions, we utilized the immunocompromised SCID and Rag^{-/-} mouse models (Figure 5H and Figure 5J). We administered TMP and doxy supplemented chow from days 1 to 14 in both mouse models, switched to chow without supplements, then switched back to supplemented chow from days 98 to 168 in the SCID model and days 56 to 126 in the Rag^{-/-} model. The TKS strain established infection with 5x10³ pulmonary CFU in SCID mice and 2.5x10³ CFU in Rag^{-/-} mice by day 14. Upon depleting TMP and doxy, bacteria cleared at a rate of ~1.5 log CFU per week in the SCID model and 0.9 log CFU per week in the Rag^{-/-} model. There was no detectable bacterial growth from lung homogenates at days 56 and 98 in the SCID model and at day 56 in the Rag^{-/-} model. After return to doxy and TMP supplementation for 10 weeks, no viable bacteria were detected at day 168 in the SCID model and day 126 in the Rag^{-/-} model in the lungs (Figures 5I and 5K). In addition, no viable bacteria were found in other organs including spleen, mediastinal lymph node, liver and bone marrow (Supplemental Figures 2 and 3). Thus, the TKS strain did not reemerge under permissive conditions, even in the absence of adaptive immunity, once it had been eliminated by kill switch induction.

ReFig. 1. The TKS strain does not relapse in the Rag^{-/-} model. (A) The scheme of relapse study in the Rag^{-/-} mice. Mice were fed with doxy/TMP chow 3 days before infection till day 14 to allow infection establish. Mice were switched back to regular chow since day 14 till day 56 for TKS clearance. The mice were put back on doxy/TMP diet for 10 weeks since day 56 to test relapse. (B to F) shows the CFU enumeration in multiple organs. The green dots indicate mice fed with permissive chow and blue dots show mice under restrictive chow. The dash line shows the CFU detection limit (5 CFU). Mean and standard deviation are shown at each time point.

2. Is it possible that individuals challenged with the triple kill strain might be immunized against the phage lysins? Perhaps its unlikely, but I wonder if antibodies against phage lysins could impair the lysis of extracellular bacteria. Although not required to support the authors conclusions about safety, it would be good to know if antibody responses against the lysins were detected in the macaques.

Reply: We thank the reviewer's insight however we are limited by resource to directly test antibody responses against the lysins in macaques. We did not bank macaque serum for serology study to address this question. This question could be answered by future studies. It is reassuring to note that mycobacterial phage lysins are remarkably diverse and there is unlikely to be much cross immunoreactivity.

Reviewer #2:

The study explores the development of an M. tuberculosis strain for potential use in human challenge experiments. The authors investigate multiple kill switches regulated by exogenous compounds

tetracycline and trimethoprim individually and in combination. The strains demonstrate immunogenicity and antibiotic susceptibility similar to wild-type Mtb under permissive conditions. They exhibit rapid elimination in the absence of external compounds, suggesting potential safety and effectiveness for a human challenge model. Notably, the three-kill-switch strain shows promising characteristics, including a minimal escape rate and no relapse in a mouse model. The authors display meticulous attention to detail and creativity in constructing kill switches, underscoring their commitment to precision in experimental design.

Comments:

1. Line 248: The others set their goal for an escape rate of $\leq 10^{-12}$. How did the authors arrive at this rate? Providing context for this goal within this magnitude will help the reader appreciate the achieved calculated mutation rate of $\leq 10^{-15}$.

Reply: In the TKS strain, the escape rates are $\sim 10^{-8}$ to 10^{-7} per genome per generation for both the ddTMP degron kill switch and dual-lysin kill switch. Given the independent mechanisms of two kill switches, the 10^{-15} theoretical combined escape rate is calculated as a product of escape rate of each individual kill switch. We set the goal of an escape rate of $\leq 10^{-12}$ due to practical reasons for experiment validation. It takes 10 L Mtb culture to measure an escape rate at a magnitude of 10^{-12} per genome per generation via fluctuation test. That high volume is at the limit of our BL3 safety capacity.

2. The mutations of the dual lysin mutants should also be visually represented similarly to the mutants of the TMP kill switch in Figure 3F.

Reply: We visualized the escape mutations in dual lysin as the reviewer suggested (**ReFig. 2**). This figure is also incorporated into the manuscript as the Figure 1D. Accordingly, we added this text in line 101-107: *We mapped escape mutations to both lysin kill switch elements via whole genome sequencing. In the L5L kill switch, most mutations were mapped to the tetO promoter region and lysin A. This indicates the L5L escape arose from either disruption of the revTetR regulation or compromised lysin A function. In the D29L kill switch, frameshift and deletion mutations were found in tetR-tetO elements, indicating that escape of D29L kill switch is likely due to alteration in TetR function (Figure 1D and Supplemental Table 1).*

ReFig. 2. The mutation density map of dual-lysin escape mutants. The escape density plot shows mutations in L5 lysin kill switch and D29 lysin kill switch from 20 independent escape mutants. Mutation type and frequency in each individual kill switch are noted.

3. Line 127: The reference used (reference 23) seems unsuitable for this statement, as the study by Forgacs et al tested clinical Mtb isolates against TMP in combination with sulfamethoxazole and detected susceptibility.

Reply: We appreciate the reviewer's comment on reference. We've updated the reference to reflect the change. Suling et al., initially reported the MIC of TMP in H37Ra above 128 $\mu\text{g/mL}$. Macingwana et al., reported that TMP had negligible inhibitory effect on H37Rv growth up to 38 $\mu\text{g/mL}$, and it reached only 44% growth inhibition at 152 $\mu\text{g/mL}$. This is consistent with our observation of no growth arrest at 50 μM (14.5 $\mu\text{g/mL}$).

4. Lines 103-114: It should be stated in the text (and Fig. 1) how much time passed between the last doxy supplementation and the necropsies of the animals to allow the reader to put those experiments into perspective. Only the method section suggests it might have been 6 weeks after the 2 weeks and 6 weeks challenge.

Reply: We appreciate the reviewer's comment and have added the following text in the main text to clarify the experiment procedure (line 112 to 115): "The NHPs were kept on either a daily doxy regimen for 6 weeks post Mtb challenge or a regimen of 2 weeks of daily doxy followed by 4 weeks of a regular diet post Mtb challenge. All 6 NHPs were subjected to necropsies at week 6 post challenge to evaluate bacterial burden." An updated NHP study scheme (**RevFig. 3**) has been added as a new Fig.1E.

ReFig. 3. The scheme of NHP study with the Mtb dual-lysin strain. 6 macaques were challenged with 6 CFU Mtb dual-lysin Erdman strain. Two macaques were fed with doxycycline-containing diet for 6 weeks, and four macaques were fed with doxycycline for 2 weeks followed by regular diet for 4 weeks. Pulmonary PET-CT was performed at week 2, 4 and 6. Comprehensive necropsies were performed at week 6 post infection.

5. Line 452: The method section of the NHP infection study is missing the quantity of administered Doxycycline.

Reply: The doxycycline was administered at 40 mg/kg PO SID for indicated duration. We have added the text in the method section.

6. The outstandingly low escape rate of 10^{-12} of the TKS strain is achieved by the combination of Doxy and TMP kill switches. Each kill switch by itself does not fulfill the strict safety guidelines. The possibility of unevenly distributed inducers in the host and complex environments of Mtb infection sites could be discussed. Exposure of the TKS strain to only one inducer could allow the strain to escape stepwise, or this might not be possible because of sufficiently high concentrations of Doxy and TMP.

Reply: Previous PK/PD studies indicate TMP and Doxy showing high tissue penetration sufficient to support the TKS strain growth as evidenced below:

Feeding C57/BL6 mice with 2000 ppm doxy in diet results in doxy plasma level at 757-1850 ng/mL at steady state¹. In this mouse model, the pulmonary doxy concentration was 2- to 4- fold higher in plasma¹. The doxy levels found in cellular and necrotic lesions are higher than in plasma in a rabbit TB infection model¹. At these doxy doses, *in vivo* levels are above the growth required doxy concentration (500 ng/mL).

Oral dosing of TMP at 4 mg/kg in Rhesus macaques leads to plasma concentration at 130-300 ng/mL², and macaque lung had 2x TMP accumulation compared to plasma^{3,4}. In mouse and rat models, pulmonary TMP concentration is 2.0-17.5x higher than serum level⁴. All these levels are higher than the TMP minimum growth required concentration (30 ng/mL).

The elimination half-life of doxycycline is 16-22 hours and trimethoprim is 8-10 hours in plasma. There could be a transient period, likely a few hours, between each dose in which only one supplement is above growth permissive concentration in plasma. We do not have the data speaking to the half-life of supplements in human tissue.

Based on reasons above, we added the following text in line 270-275: “*We reasoned that it is unlikely for the TKS strain to develop stepwise escape mutations in each kill switch sequentially from uneven tissue distribution of doxy and TMP. Previous PK/PD studies with doxy and TMP indicate good oral absorbance, high serum concentrations and excellent tissue penetration^{1, 2, 3, 4}. The pulmonary levels of both drugs are above the minimum growth required concentration for the TKS strain.*”

For consistency, please identify the dashed line in Fig. 1B/E, Fig. 5B/C/I.

Reply: We have added “*The dashed line in CFU panels indicates the CFU lower detection limit.*” in the legends of Fig. 1 and Fig. 5.

Indicate the organism in Fig. 5E similarly to panels A-D.

Reply: We have clarified the organism as C57BL/6 mouse in the legend of Fig. 5D, Fig. 5E and Fig. 5F.

Fig. 5 text: Please consistently name the strains. The “combo kill switch strain” should be the TKS strain.

Reply: We have substituted “The combo kill switch strain” to “The TKS strain” in the main text and legends.

Reviewer #3:

This manuscript by Wang et al describes generation and characterization of a strain of Mtb containing three kill switches. The investigators first generated a tetracycline-dependent dual lysin Mtb strain and then engineered a strain to combine two lysin switches with NadE-ddTMP to yield a triple-kill switch (TKS) Mtb H37Rv strain. The investigators characterize growth kinetics, antibiotic susceptibility, and immune profiling of the TKS strain in mice. The experiments described are well controlled and of high rigor and create a strong foundation for future studies to continue characterization of the TKS strain in evaluating its potential utility for development and application in human infection model (CHIM) studies.

The studies are of high importance given the current lack of an effective vaccine against pulmonary TB in adolescents and adults and lack of well-defined immune correlates of protection against TB. Development of a safe controlled human infection model (CHIM) for Mtb would be advantageous to the field of TB vaccinology and treatment. The investigators use SCID mice to demonstrate lack of growth of the TKS strain in lung and spleen after return to doxy and TMP supplementation. Rigorous safety studies will need to be conducted prior to use of engineered Mtb strains for use in CHIM studies, and the SCID experiments presented in this manuscript provide important preliminary data towards establishing a safety profile of this strain for potential use in humans. Clarification on the comments below would strengthen the conclusions that can be drawn from the manuscript.

Comments:

1. The Abstract (lines 31-32) and conclusion of the Discussion section (lines 281-282) stating that the immunogenicity of TKS strain is comparable to the wild-type strain is not strongly supported by the data presented in this manuscript. Minimal immune profiling data are shown (only one time point was evaluated, with no evaluation of either innate immune responses or Mtb-specific T cell responses). Immune profiling data of TKS-infected mice are presented along with uninfected mice, although there are no immune profiling data comparing TKS with wt Mtb. Either the Abstract/Discussion summary and conclusions should be modified to more accurately reflect the data presented in the manuscript, or additional experiments provided to demonstrate the direct comparison of immunogenicity of TKS with wt Mtb.

Reply: We agree with the reviewer's comment on immunogenicity of the TKS strain and understanding this fully will require future study. We profiled T cells at day 84 in C57BL/6 mice to characterize the duration of the adaptive response after clearance of the TKS strain. To accurately reflect the current data, we modified the line 32 and 33 in abstract as "*The resultant strains were able to infect mouse models and displayed antibiotic susceptibility similar to wild-type Mtb under permissive conditions.*", and modified the line 294 to 296 in discussion as "*Future TB CHIM work using the attenuated Mtb TKS strain could compare the immunogenicity between the TKS strain and wild-type Mtb strain infection, and illuminate host-pathogen interaction more relevant to Mtb infection than the BCG and PPD model.*".

2. Figure 5, panels D-G: it is not clear what condition is represented by the grey bars. The implication from the Results section (line 217) is that the grey bars are uninfected mice, although this should be clarified in Figure 5 and in the figure legend.

Reply: We thank the reviewer's suggestion and have clarified in Figure 5 and its legend.

3. Lung CD4 and CD8 T cells were evaluated at day 84 post-infection with the TKS strain. T cells were evaluated for activation and effector T cell populations, although Mtb-specific immunogenicity was not evaluated. Only one time point was evaluated for immunogenicity (d84), although it would be useful to evaluate immunogenicity at earlier time points, as well as comparison with mice infected with wt H37Rv.

Reply: We thank the reviewer's suggestion. We agree that Mtb-specific T cell response has not been evaluated in this paper, and future study will elucidate the TKS strain-induced immune response and its immunogenicity. To accurately reflect the data, we have made changes in line 32 and 33 of the abstract as *"The resultant strains were able to infect mouse models and displayed antibiotic susceptibility similar to wild-type Mtb under permissive conditions."*, and modified the line 294 to 296 in discussion as *"Future TB CHIM work using the attenuated Mtb TKS strain could compare the immunogenicity between the TKS strain and wild-type Mtb strain infection, and illuminate host-pathogen interaction more relevant to Mtb infection than the BCG and PPD model."*

4. Line 217 indicates the T cell response at d84 in mice under restrictive conditions were distinguishable from uninfected mice, although this conclusion is not strongly supported by the data in Figure 5D-E. There are no differences in CD4 T cell responses between uninfected and mice under restrictive conditions (Figure 5D, E) and only modestly higher frequencies of effector CD8 T cell responses under restrictive conditions, compared with uninfected mice (Figure 5G). This conclusion should be modified to reflect the data in Figure 5 more accurately.

Reply: We agree with the reviewer's comment. We have added the following text in line 226 to 230 to accurately reflect the data we observed: *"Within the T cell population expressing markers above, we observed significantly attenuated CD4 and CD8 responses under restrictive conditions compared to permissive conditions. After the TKS strain clearance, the CD4 response is indistinguishable from uninfected controls. Similarly, after the TKS strain clearance, the CD8 response is largely similar to uninfected control at day 84, except for a slightly increased proportion of CD8⁺ effector T cells among the total CD8 T cell population."*

5. In experiments of TKS infection of SCID mice, lung and spleen were cultured for CFU, but no other tissues were cultured. Viable bacteria may be present in either LN, BM, or other tissues. The number of SCID mice evaluated was small (n=5 for each time point), and although the experiments performed provide promising data, it will be important to demonstrate eradication of viable bacteria in a more

comprehensive manner and in a larger number of animals in future studies characterizing the TKS strain prior to use in humans.

Reply: Thank you. As described above, we have performed a new experiment using Rag^{-/-} mice. The new data and discussion are addressed in the reviewer #1's comment #1. The new Rag^{-/-} data is also incorporated as Figures 5J and 5K.

6. Infection of SCID mice is important to demonstrate lack of reemergence of bacterial growth, although the number of SCID mice per time point is small (n=5) and warrants repeating with larger numbers in future studies (not necessarily in this manuscript) to more robustly characterize the TKS strain prior to use in CHIM studies. The authors could consider expanding on this point in the Discussion section.

Reply: We agree with the reviewer's suggestion. We have repeated the relapse study in Rag^{-/-} mouse model, and the data is incorporated in Figure 5. We also expanded the discussion in line 297 to 300 as the following text: *In animal experiments, bacteria grew and cleared in a manner similar to what we observed in vitro. However, the kinetics of growth and clearance could well differ in other species and these results certainly do not perfectly predict what would occur in human infection.*

7. The TKS strain requires presence of both aTc and TMP, provided to mice in the form of supplemented chow. The authors suggest that the TKS strain could be used in human challenge studies, although the feasibility and practicality of conducting PK/PD studies to establish necessary levels of aTc and TMP in humans to ensure safety of the TKS strain in human challenge studies is not clear and could be further acknowledged as a limitation in the Discussion section.

Reply: We thank the reviewer's comment on aTc and TMP human PK/PD. In the animal study, we replaced aTc with doxycycline (doxy) due to higher *in vivo* absorption and stability. Doxycycline and trimethoprim have been widely used as antibiotics in human, and previous literature indicates both compounds showing good oral absorption, high serum concentrations and excellent tissue penetration^{1, 2, 3, 4}. The reviewer #2 raised a similar comment (comment #6) and we have addressed the related concern above.

Reference

1. Gengenbacher M, *et al.* Tissue Distribution of Doxycycline in Animal Models of Tuberculosis. *Antimicrob Agents Chemother* **64**, (2020).
2. Hobbs CV, *et al.* HIV treatments reduce malaria liver stage burden in a non-human primate model of malaria infection at clinically relevant concentrations in vivo. *PLoS One* **9**, e100138 (2014).
3. Craig WA, Kunin CM. Distribution of trimethoprim-sulfamethoxazole in tissues of rhesus monkeys. *J Infect Dis* **128**, Suppl:575-579 p (1973).

4. Patel RB, Welling PG. Clinical pharmacokinetics of co-trimoxazole (trimethoprim-sulphamethoxazole). *Clin Pharmacokinet* **5**, 405-423 (1980).

Response to Reviewers

We thank the reviewers for the thorough assessment of our manuscript and the insightful feedback. We have performed additional experiments and analyses in order to address these comments. Below we included responses to each point by the reviewers in blue.

Reviewer #1:

The new data in the Rag^{-/-} mice has nicely confirmed the previous results, and all of my concerns have been addressed.

Reply: We thank the reviewer's comment.

Reviewer #2:

Thank you for the detailed response to the review comments. I appreciate the effort taken to address all the raised questions. The rebuttal document has sufficiently clarified the issues, and I am satisfied with the answers provided.

Reply: We thank the reviewer's comment.

I acknowledge that the authors' internal goal of achieving a mutation rate with an escape rate of $\leq 10^{-12}$ was influenced by technical limitations. Although this was not fully clarified in the manuscript, the published rebuttal document will provide an adequate explanation.

Reply: To clarify the rationale of expected escape rate of $\leq 10^{-12}$, we have added the explanation to lines 273 to 276 in the manuscript.

I noticed a slight inconsistency in how data points below the limit of detection are represented. In Figure 1, the data points are placed directly on the "limit of detection" line, while in Figures 4 and 5, some points fall below it. If both cases are intended to convey that no bacteria were detectable, I would suggest opting for a consistent representation.

Reply: We thank the reviewer's suggestion. We have edited figures, specifically figures 1F, 4G, 5C, 5I and 5K, to achieve consistency of data presentation.

Reviewer #3:

The authors have carefully and thoughtfully addressed each of the comments raised in the initial review of this manuscript. The authors have repeated the experiments with Rag^{-/-} mice, which is a further strength of the revised manuscript. Well done to the authors on an excellent study. I have no further comments or concerns to be addressed.

Reply: We thank the reviewer's comment.